# Micro-Scale Spherical and Cylindrical Surface Modeling via Metaheuristic Algorithms and Micro Laser Line Projection

**J. Apolinar Muñoz Rodríguez**

Centro de Investigaciones en Óptica, A. C., Lomas del Bosque 115, Col. Comas del Campestre, Leon 37000, GTO, Mexico; munoza@cio.mx

**Abstract:** With the increasing micro-scale manufacturing industry, the micro-scale spherical and cylindrical surface modeling has become an important factor in the manufacturing process. Thus, the micro-scale manufacturing processes require efficient micro-scale spherical and cylindrical models to achieve accurate assembly. Therefore, it is necessary to implement models to represent micro-scale spherical and cylindrical surfaces. This study addresses metaheuristic algorithms based on micro laser line projection to perform micro-scale spherical and cylindrical surface modeling. In this technique, the micro-scale surface is recovered by an optical microscope system, which computes the surface coordinates via micro laser line projection. From the surface coordinates, a genetic algorithm determines the parameters of the mathematical models to represent the spherical and cylindrical surfaces. The genetic algorithm performs exploration and exploitation in the search space to optimize the models' mathematical parameters. The search space is constructed via surface data to provide the optimal parameters, which determine the spherical and cylindrical surface models. The proposed technique improves the fitting accuracy of the micro-scale spherical and cylindrical surface modeling performed via optical microscope systems. This contribution is elucidated by a discussion about the model fitting between the genetic algorithms based on micro laser line projection and the optical microscope systems.

**Keywords:** micro-scale spherical and cylindrical surface modeling; genetic algorithms; micro laser line projection; optical microscope vision system

## 1. Introduction

Nowadays, the micro-scale spherical and cylindrical surface modeling plays an important role in inspecting micro-scale sphericity and cylindricity [1,2]. In the micro-scale manufacturing industry, object machining, surface roughness, drilling, and object measurements are determined via spherical and cylindrical surface models and robotic vision systems [3–5]. To achieve these manufacturing processes, mathematical models have been implemented to represent micro-scale spherical and cylindrical surfaces. For instance, the spherical surface modeling has been implemented in geomatics to determine rock surface sphericity [6,7], in optics to estimate lens sphericity [8,9], in object machining to determine surface sphericity [10,11], in pharmaceutics to determine proppants sphericity [12,13], in object prototyping to determine assemble sphericity [14,15], and so on. Moreover, the cylindrical surface modeling has been implemented in surface machining to inspect cylindrical surface [16,17], in surface milling to estimate milling cylindricity [18], and so on. Based on these statements, surface models have been constructed via surface data and mathematical equations for representing micro-scale spherical and cylindrical surfaces [19,20]. Currently, micro-scale spherical and cylindrical surface models are carried out via surface information, which is retrieved by means of optical microscope imaging systems [21,22]. These microscope systems construct the surface models by employing surface data obtained via gray-level image processing and the least squares method [23,24]. In addition, the least squares optimization is employed by the vision systems' models to determine surface

data [25]. However, sometimes the least squared method does not provide the most accurate model fitting [26]. Therefore, artificial intelligence algorithms have been implemented to accurately construct surface models. For instance, an algorithm with an adaptive fuzzy logic controller has been implemented to construct spherical surface models [27]. This algorithm minimizes an error function by employing surface data to obtain the spherical surface model parameters. Furthermore, a harmony search algorithm has been implemented to construct cylindrical surface models [28]. This algorithm implements a music process by employing surface data to optimize an error function to obtain the cylindrical surface model parameters. Usually, the spherical and cylindrical surface modeling is carried out by an optical microscope imaging system, which retrieves the surface shape via gray-level image processing. For instance, an optical microscope system based on machine vision determines micro-scale spherical surface via gray-level image processing [29]. In the same way, an optical microscope imaging system computes the micro-scale cylindrical surface by means of a frequency transform via a gray-level image [30]. The above-mentioned optical microscope systems determine the micro-scale surface topography via a gray-level profile. However, the gray-level profile does not depict accurately the surface profile. It is because the gray-level depends on the light source, surface reflectance, and the viewing angle. This means that the gray-level profile does not reproduce the surface topography. Therefore, the spherical and cylindrical surface models do not fit to the surface with great accuracy. This leads to a decrease in the micro-scale surface modeling accuracy. Based on these statements, it is established that the micro-scale surface modeling performed by the optical microscope imaging systems still represents a difficult task. To improve the micro-scale spherical and cylindrical surface modeling accuracy, the surface models should be generated by means of the surface topography.

The proposed micro-scale spherical and cylindrical surface modeling is carried out by a metaheuristic genetic algorithm based on micro laser line projection, which depicts accurately the surface topography. The metaheuristic algorithms include algorithms such as genetic algorithms, particle swarm optimization, ant colony optimization, and simulated annealing [31]. Metaheuristic algorithms such as particle swarm optimization, ant colony optimization, and simulated annealing have been implemented to construct mathematical models to represent the free-form surface [32,33]. For instance, the particle swarm makes the optimization via the position and velocity of particles [34]. Where, the objective function is provided by an inertia equation and the population is determined by a particle velocity equation. The ant colony optimization chooses paths marked by a strong pheromone concentration to determine the surface model parameters [35]. The simulated annealing makes a random search, which decreases and increases the objective function to determine the surface model [36], where the tentative solution is generated by a small perturbation. Moreover, the metaheuristic algorithms construct models via multi-objective optimization in domains such as online learning, scheduling, transportation, medicine, data classification, and so on. For instance, the online learning optimizes many objective functions to determine the mathematical models [37]. In addition, the multi-objective optimization via evolutionary algorithms is implemented to determine the parameters of mathematical models [38]. On the other hand, the scheduling domain employs structures of different metaheuristic algorithms to determine the parameters of mathematical models [39]. Additionally, the transportation domain employs several metaheuristic algorithms to determine routing of mathematical models [40]. Additionally, the evacuation domain employs several metaheuristic algorithms to optimize the multi-objective functions to determine routing of mathematical models [41]. Furthermore, the metaheuristic algorithms employ data classification to determine diagnoses in medicine [42]. The above-mentioned algorithms optimize parameters for mathematical models, which are not defined by a specific equation. For instance, the particle swarm, ant colony, and simulated annealing construct free-form surface models, which are not defined by a specific equation. Furthermore, the metaheuristic algorithms based on multi-objective optimization determine mathematical models, which are not defined by a specific equation. This leads to implementing complex

algorithms due to the missing of a reference equation to optimize the model parameters. Moreover, these algorithms begin the optimization with a random solution, which leads to computing a huge iterations number. Furthermore, the objective function is deduced by an equation, which includes additional parameters to the surface model. Furthermore, the spherical and cylindrical surfaces can be represented by having a specific equation. In this way, a genetic algorithm provides a suitable structure to optimize the spherical and cylindrical surface models. For instance, the genetic algorithm is allowed to define an objective function by means of the equations that represent the spherical and cylindrical surface. Therefore, additional parameters are not required. In addition, it is possible to begin the optimization by employing the best candidates, which provides a solution near the optimal solution. The best candidates are deduced from the search space via spherical and cylindrical equations and known surface data. Moreover, exploration and exploitation are carried out to find the optimal solution inside or outside of the best candidates. Thus, all research space can be analyzed. Based on these statements, the metaheuristic genetic algorithm is chosen to construct the spherical and cylindrical surface models. The genetic algorithm viability to construct micro-scale spherical and cylindrical surface models is elucidated via quality gap, solution quality, suitable structure, and experimental results. Thus, the genetic algorithm determines the spherical and cylindrical modeling by employing surface coordinates, which are computed via micro laser line projection. To do so, the metaheuristic algorithm determines the search space from the surface topography to compute the surface model parameters. In this procedure, the genetic algorithm determines the best candidates for the initial population via analytic equations and surface data. Then, the algorithm performs an optimization via exploration and exploitation to find the optimal surface model parameters. The genetic algorithm based on micro laser line projection performs micro-scale spherical and cylindrical surface modeling for rectangular surfaces. This micro-scale surface modeling is carried out by means of an optical microscope vision system on which a CCD camera and a 42 μm laser line are attached. In this way, the micro laser line is projected on the surface and the CCD camera captures the micro laser line, whose reflection depicts the surface topography. The surface dimension is computed via perspective model and the laser line coordinates. Thus, the micro-scale spherical and cylindrical surface modeling is performed via surface topography, which is not employed by the traditional optical microscope imaging systems. Therefore, the micro-scale surface modeling via genetic algorithms based on micro laser line projection improves the fitting accuracy of the micro-scale spherical and cylindrical surface modeling performed via optical microscope systems. This statement is corroborated by a discussion based on model fitting of the micro-scale spherical and cylindrical surface modeling. Thus, the contribution of the proposed technique is elucidated via micro-scale surface modeling accuracy. The accuracy is determined via the error between the surface model provided by the genetic algorithm and the surface data. The paper is organized as follows: the algorithm to perform the micro-scale spherical surface modeling is described in Section 2.1, the algorithm to perform the micro-scale cylindrical surface modeling is described in Section 2.2, the micro-scale surface recovering via microscope vision system based on micro laser line projection is described in Section 2.3, the results of the micro-scale spherical and cylindrical surface modeling are pointed out in Section 3, and a discussion about the model fitting is described in Section 4.

## 2. Materials and Methods

### 2.1. Micro-Scale Spherical Surface Modeling via Genetic Algorithm

The micro-scale spherical surface modeling is carried out by a genetic algorithm via surface coordinates, which are retrieved via micro laser line projection from a rectangular spherical surface. Typically, a metaheuristic algorithm provides an optimization in a moderated time [43]. Additionally, the metaheuristic algorithm performs an exploration and exploitation to find the optimal solution [44]. Based on these statements, the micro-scale spherical surface modeling is performed by a genetic algorithm by means of the surface

coordinates provided by a micro laser line image. To do so, a spherical surface model is constructed by means of surface coordinates, which are retrieved from a rectangular spherical surface. The surface coordinates are shown in Figure 1. These surface coordinates are represented by $(x_{i,j}, y_{i,j}, z_{i,j})$, where the sub-indices $(i, j)$ are defined in the $x$-axis and $y$-axis, respectively. Thus, the rectangular spherical surface provides the surface points $(z_{0,0}, z_{1,0}, \ldots, z_{n,0}, z_{n,1}, \ldots, z_{n,m})$ in the $z$-axis, where, the sub-indices $(n, m)$ depict the number of surface points in the $x$-axis and $y$-axis, respectively.

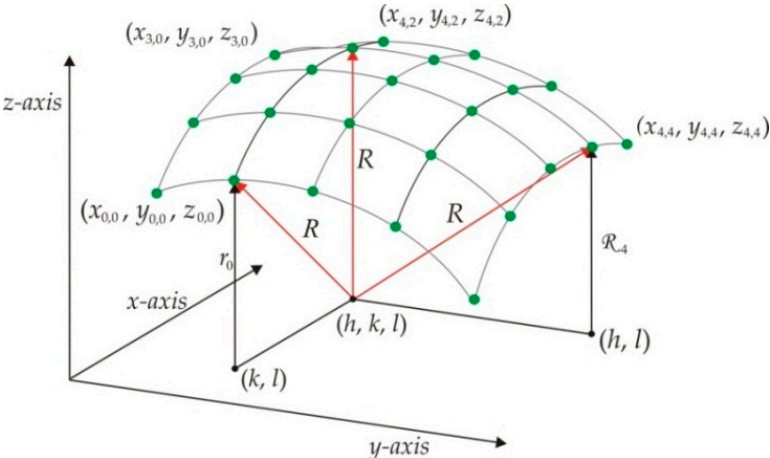

**Figure 1.** Rectangular spherical surface to perform spherical surface modeling.

The spherical origin coordinates are represented by $(h, k, l)$ and the ratio is represented by $R$. Furthermore, the spherical surface provides the ratio $r_i$ in the $x$-axis and the ratio $\mathcal{R}_j$ in the $y$-axis. The coordinates $(k, l)$ and $(h, l)$ represent the origin coordinates of the ratios $r_i$ and $\mathcal{R}_j$, respectively. Based on these coordinates, the spherical surface model is defined via ratio $R$ by means of the next expression

$$(x_{i,j} - h)^2 + (y_{i,j} - k)^2 + (z_{i,j} - l)^2 = R^2 \tag{1}$$

Additionally, the circumference equation is defined via ratio $r_i$ and $\mathcal{R}_j$ by the following expressions

$$(y_{i,j} - k)^2 + (z_{i,j} - l)^2 = r_i^2 \tag{2}$$

$$(x_{i,j} - h)^2 + (z_{i,j} - l)^2 = \mathcal{R}_j^2 \tag{3}$$

From these equations, the spherical surface model is deduced by computing the origin coordinates $(h, k, l)$ and the ratio $R$ via genetic algorithm. To do so, a genetic algorithm is implemented via Equations (1)–(3) and the surface data $(x_{i,j}, y_{i,j}, z_{i,j})$. Thus, the genetic algorithm determines the spherical parameters in five steps, which are described as follows.

The first step is to determine the initial population. Typically, the metaheuristic algorithms determine the initial population in random form. In addition, the initial population can be determined by means of the best candidates. This leads to a speed out of the convergence [45]. The best candidates can be obtained via random methods or by sampling the surface model. Therefore, the genetic algorithm determines the initial population by sampling the surface model to obtain the best candidates. The sampling is carried out by solving Equations (2) and (3) to obtain the best candidates. Based on these statements, Equations (2) and (3) are solved by employing the surface coordinates $(x_{i,j}, y_{i,j}, z_{i,j})$, $(x_{i+\delta,j}, y_{i+\delta,j}, z_{i+\delta,j})$, $(x_{i,j+\Delta}, y_{i,j+\Delta}, z_{i,j+\Delta})$, and $(x_{i+\delta,j+\Delta}, y_{i+\delta,j+\Delta}, z_{i+\delta,j+\Delta})$. In this case, the sub-index $\Delta$ indicates

an increment such as $\Delta = m/2$ and the sub-index $\delta$ indicates an increment such as $\delta = n/2$. Thus, the initial parameters $(h, k, l)$ are determined by means of the next expressions

$$k = \frac{\mathcal{Z}_1[\mathcal{Y}_4 + \mathcal{Z}_4] - \mathcal{Z}_3[\mathcal{Y}_2 + \mathcal{Z}_2]}{2[\mathcal{Y}_3 \mathcal{Z}_1 - \mathcal{Y}_1 \mathcal{Z}_3]} \tag{4}$$

$$l = \frac{\mathcal{Y}_1[\mathcal{Y}_4 + \mathcal{Z}_4] - \mathcal{Y}_3[\mathcal{Y}_2 + \mathcal{Z}_2]}{2[\mathcal{Z}_3 \mathcal{Y}_1 - \mathcal{Z}_1 \mathcal{Y}_3]} \tag{5}$$

$$h = \frac{\mathcal{Z}_5[\mathcal{X}_8 + \mathcal{Z}_8] - \mathcal{Z}_7[\mathcal{X}_6 + \mathcal{Z}_6]}{2[\mathcal{X}_7 \mathcal{Z}_5 - \mathcal{X}_5 \mathcal{Z}_7]} \tag{6}$$

where, $\mathcal{Z}_1 = (z_{i,j+\Delta}) - (z_{i,j})$, $\mathcal{Z}_2 = (z_{i,j+\Delta})^2 - (z_{i,j})^2$, $\mathcal{Z}_3 = (z_{i+\delta,j+\Delta}) - (z_{i+\delta,j})$, $\mathcal{Z}_4 = (z_{i+\delta,j+\Delta})^2 - (z_{i+\delta,j})^2$, $\mathcal{Z}_5 = (z_{i+\delta,j}) - (z_{i,j})$, $\mathcal{Z}_6 = (z_{i+\delta,j})^2 - (z_{i,j})^2$, $\mathcal{Z}_7 = (z_{i+\delta,j+\Delta}) - (z_{i,j+\Delta})$, $\mathcal{Z}_8 = (z_{i+\delta,j+\Delta})^2 - (z_{i,j+\Delta})^2$, $\mathcal{Y}_1 = (y_{i,j+\Delta}) - (y_{i,j})$, $\mathcal{Y}_2 = (y_{i,j+\Delta})^2 - (y_{i,j})^2$, $\mathcal{Y}_3 = (y_{i+\delta,j+\Delta}) - (y_{i+\delta,j})$, $\mathcal{Y}_4 = (y_{i+\delta,j+\Delta})^2 - (y_{i+\delta,j})^2$, $\mathcal{X}_5 = (x_{i+\delta,j}) - (x_{i,j})$, $\mathcal{X}_6 = (x_{i+\delta,j})^2 - (x_{i,j})^2$, $\mathcal{X}_7 = (x_{i+\delta,j+\Delta}) - (x_{i,j+\Delta})$, and $\mathcal{X}_8 = (x_{i+\delta,j+\Delta})^2 - (x_{i,j+\Delta})^2$. By computing Equations (4)–(6), four parents $(P_{1,t}, P_{2,t}, P_{3,t}, P_{4,t})$ are determined for each parameter, where, the sub $-$ index $t$ indicates the generation number. In this way, the parent $P_{1,1}$ is determined by computing Equations (4)–(6) via sub-indices $(i = 0, j = 0, \delta = n/2, \Delta = m/2)$. The parent $P_{2,1}$ is determined by computing Equations (4)–(6) via sub-indices $(i = 0, j = m/2, \delta = n/2, \Delta = m/2)$. The parent $P_{3,1}$ is determined by computing Equations (4)–(6) via sub-indices $(i = n/2, j = 0, \delta = n/2, \Delta = m/2)$. The parent $P_{4,1}$ is determined by computing Equations (4)–(6) via sub-indices $(i = n/2, j = m/2, \delta = n/2, \Delta = m/2)$. Thus, four coordinates $(h, k, l)$ are obtained, and they are defined as the initial population. Additionally, the ratio $\check{R}$ is determined by the average distance between the origin coordinates and the surface points. Then, the genetic algorithm determines the search space based on the maximum and minimum value of each parameter. In this way, the minimum and maximum of the parameter $h$ are defined as $x_{0,0}$ and $x_{n,m}$, respectively. The minimum and maximum of the parameter $k$ are defined as $y_{0,0}$ and $y_{n,m}$, respectively. The minimum and maximum of the parameter $l$ are defined as $(l + \check{R}/2)$ and $(l - \check{R}/2)$, respectively. Thus, the initial population is completed.

The second step is to perform the crossover to create the current children. This procedure is carried out via exploration and exploitation [46]. In this procedure, two children are created inside parents and two children are created outside parents. Thus, the children $(C_{1+4*q,t}, C_{2+4*q,t}, C_{3+4*q,t}, C_{4+4*q,t})$ are created via parents $(P_{1+2*q,t}, P_{2+2*q,t})$ for $q = 0$ and $q = 1$. These children are computed by means of the next equations

$$C_{1+4*q,t} = \begin{cases} P_{1+2*q,t} - 0.5\beta \left| P_{1+2*q,t} - \text{minimum} \right|, & \text{if } P_{1+2*q,t} < P_{2+2*q,t} \\ P_{2+2*q,t} - 0.5\beta \left| P_{2+2*q,t} - \text{minimum} \right|, & \text{if } P_{2+2*q,t} < P_{1+2*q,t} \end{cases} \tag{7}$$

$$C_{2+4*q,t} = 0.5\left[ \left( P_{1+2*q,t} + P_{2+2*q,t} \right) - \beta \left| P_{1+2*q,t} - P_{2+2*q,t} \right| \right] \tag{8}$$

$$C_{3+4*q,t} = 0.5\left[ \left( P_{1+2*q,t} + P_{2+2*q,t} \right) + \beta \left| P_{1+2*q,t} - P_{2+2*q,t} \right| \right] \tag{9}$$

$$C_{4+4*q,t} = \begin{cases} P_{2+2*q,t} + 0.5\beta \left| \text{maximum} - P_{2+2*q,t} \right|, & \text{if } P_{1+2*q,t} < P_{2+2*q,t} \\ P_{1+2*q,t} + 0.5\beta \left| \text{maximum} - P_{1+2*q,t} \right|, & \text{if } P_{2+2*q,t} < P_{1+2*q,t} \end{cases} \tag{10}$$

For these equations, the probability distribution $\beta$ is accomplished via spread factor $\alpha$, which is randomly generated in the interval between 0 and 1. Thus, the probability distribution is computed by the expression $\beta = (2\alpha)^{1/2}$ if $\alpha > 0.5$, otherwise, $\beta = [2(1 - \alpha)]^{1/2}$. In this way, Equations (8) and (9) compute the children inside parents, and Equations (7) and (10) compute the children outside parents. By replacing the parents $(P_{1,t}, P_{2,t})$ and $q = 0$ in Equations (7)–(10), the children $(C_{1,t}, C_{2,t}, C_{3,t}, C_{4,t})$ are determined. In addition, by replacing the parents $(P_{3,t}, P_{4,t})$ and $q = 1$ in Equations (7)–(10), the children $(C_{5,t}, C_{6,t}, C_{7,t}, C_{8,t})$ are determined.

The third step is to evaluate the fitness. To do so, an objective function is constructed based on the surface coordinates $(x_{i,j}, y_{i,j}, z_{i,j})$ and the current origin coordinates $(h, k, l)$. The objective function is defined by means of the next expression

$$FO = \min\left\{ \frac{1}{nxm} \sum_{i=0}^{n} \sum_{j=0}^{m} \left| \breve{R} - \sqrt{(x_{i,j} - h)^2 + (y_{i,j} - k)^2 + (z_{i,j} - l)^2} \right| \right\} \qquad (11)$$

$$\breve{R} = \frac{1}{nxm} \sum_{i=0}^{n} \sum_{j=0}^{m} \left[ \sqrt{(x_{i,j} - h)^2 + (y_{i,j} - k)^2 + (z_{i,j} - l)^2} \right]$$

The fourth step is the selection procedure, which determines the parents of the next generation. To do so, the best current parents and children are selected via fitness. In this way, the parents $P_{1,t+1}$ and $P_{3,t+1}$ are selected from the pairs $(P_{1,t}, P_{2,t})$ and $(P_{3,t}, P_{4,t})$, respectively. Then, the parents $P_{2,t+1}$ and $P_{4,t+1}$ are selected from the children $(C_{1,t}, C_{2,t}, C_{3,t}, C_{4,t})$ and $(C_{5,t}, C_{6,t}, C_{7,t}, C_{8,t})$, respectively.

The fifth step is to perform the mutation procedure to avoid trapping in a local minimum. To do so, the worst parent is selected by computing the fitness via Equation (11). Then, a new parent is randomly generated from the search space and the fitness is computed via Equation (11). If the new parent improves the fitness, the worst parent is replaced by the new parent. On the other hand, the worst parent is not mutated. Additionally, a parent is randomly selected to mutate one parameter, which is selected in random form. To do so, a new parameter is randomly generated from the search space. Then, the new parameter is replaced in the selected parent and the fitness is computed via Equation (11). If the parent fitness is improved, the new parameter is replaced, if not, the parameter is not mutated. Thus, the current parents' mutation is completed and the $(t + 1)$ generation parents are determined. Then, the crossover is performed to create the $(t + 1)$ generation children via Equations (7)–(10). In addition, the fitness of these children is computed via Equation (11). Thus, the population of the $(t + 1)$ generation is completed. The procedure to compute the $(t + 1)$ generation is repeated until the optimal parameters $(h, k, l)$ that minimize the objective function FO are obtained. In this way, the spherical surface modeling is completed.

To elucidate the surface modeling via genetic algorithm, a spherical surface model is constructed from the spherical surface shown in Figure 2a. The flowchart of the genetic algorithm to construct the spherical surface modeling is shown in Figure 2b. From this structure, the genetic algorithm performs the first step to determine the initial population. To do so, Equations (4)–(6) are computed by employing $m = 10$, $n = 10$, $\Delta = 5$, and $\delta = 5$. Thus, the parent $P_{1,1}$ is obtained by computing Equations (4)–(6) via surface coordinates whose sub-indexes are ($i = 0$, $j = 0$, $\delta = 5$, $\Delta = 5$), the parent $P_{2,1}$ is determined via sub-indexes ($i = 0$, $j = 5$, $\delta = 5$, $\Delta = 5$), the parent $P_{3,1}$ is deduced via sub-indexes ($i = 5$, $j = 0$, $\delta = 5$, $\Delta = 5$), and the parent $P_{4,1}$ is determined via sub-indexes ($i = 5$, $j = 5$, $\delta = 5$, $\Delta = 5$). In this way, four origin coordinates $(h, k, l)$ are obtained, and they are defined as the initial population. The initial population data are shown in Table 1, where the first column represents the origin coordinates, the second column represents the generation number, and the parents $(P_{1,1}, P_{2,1}, P_{3,1}\ P_{4,1})$ are pointed out in the third to sixth column.

Then, the genetic algorithm performs the second step via crossover to compute the current children. To do so, the children $(C_{1,t}, C_{2,t}, C_{3,t}, C_{4,t})$ are computed by substituting the parents $(P_{1+2*q,t}, P_{2+2*q,t})$ and $q = 0$ in Equations (7)–(10). Additionally, the children $(C_{5,t}, C_{6,t}, C_{7,t}, C_{8,t})$ are computed by substituting the parents $(P_{1+2*q,t}, P_{2+2*q,t})$ and $q = 1$ in Equations (7)–(10). The data of the children are pointed out in the seventh to fourteenth column of Table 1.

Then, the algorithm performs the third step to evaluate the fitness of the current generation. To do so, the coordinates $(h, k, l)$ of each parent and son are replaced in Equation (11) to compute the objective function. The current generation fitness is pointed out in the fourth row of Table 1. Thus, the first generation is completed. In this case, the

algorithm provides a low error from the initial population. Therefore, it is corroborated that the initial population provides the best candidates to speed out the convergence.

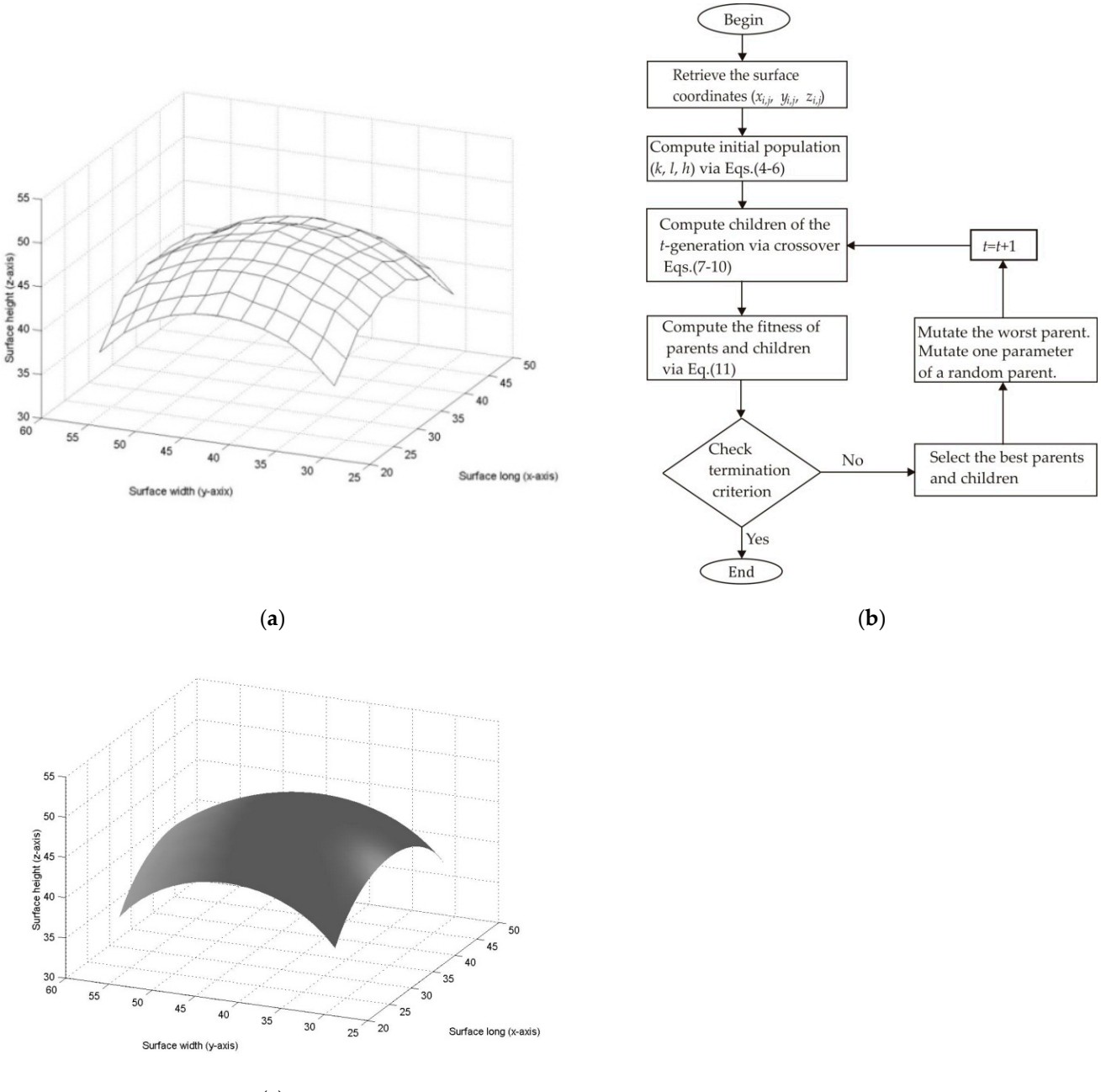

(**a**)　　　　　　　　　　　　　　　　　　　　　　(**b**)

(**c**)

**Figure 2.** (**a**) Spherical surface to perform spherical surface modeling. (**b**) Flowchart of the genetic algorithm to determine spherical surface model. (**c**) Spherical surface provided by the surface model in Equation (1).

Then, the algorithm performs the fourth step to select the next generation parents. To do so, the best current parents and children are selected via fitness. In this way, the parents $P_{1,t+1}$ and $P_{3,t+1}$ are selected from the pairs $(P_{1,t}, P_{2,t})$ and $(P_{3,t}, P_{4,t})$, respectively. In addition, the parents $P_{2,t+1}$ and $P_{4,t+1}$ are selected from the children $(C_{1,t}, C_{2,t}, C_{3,t}, C_{4,t})$, $(C_{5,t}, C_{6,t}, C_{7,t}, C_{8,t})$, respectively. In this case, $P_{1,2} = P_{2,1}$, $P_{3,3} = P_{4,1}$, $P_{2,2} = C_{2,1}$, and $P_{4,2} = C_{7,1}$.

Then, the algorithm performs the fifth step to mutate the parent $P_{2,2}$, which is determined as the worst parent via fitness. To perform the mutation, a new parent is randomly generated from the search space and the fitness is computed via Equation (11). In this case, the new parent does not improve the fitness and the parent is not mutated. Next, the parent $P_{2,3}$ is randomly selected to mutate the parameter $h$, which is selected in random form. Thus, a new parameter $h$ is randomly generated from the search space and the fitness of the parent $P_{2,3}$ is computed via Equation (11). In this case, the fitness is improved by the new parameter. Therefore, the parameter $h$ is mutated. Then, the algorithm performs the second step to create the children of the $(t + 1)$ generation via Equations (7)–(10). Additionally, the fitness of these children is computed via Equation (11). Thus, the population of the second generation is completed. The second-generation population is pointed out in the fifth to eighth row of Table 1. In this way, the procedure to compute the $(t + 1)$ generation is repeated until obtaining the optimal parameters $(h, k, l)$, which minimize the objective function Equation (11). The optimal origin coordinates are $h = 30.1214$, $k = 40.0256$, $l = 29.9715$, and $R = \check{R} = 20.2304$. These coordinates are substituted in Equation (1) to obtain the spherical surface model $z_{i,j} = l + [R^2 - (x_{i,j} - h)^2 - (y_{i,j} - k)^2]^{1/2}$, which produces the spherical surface shown in Figure 2c. Thus, the spherical surface modeling is performed via the metaheuristic algorithm. The cylindrical surface modeling is described in the next section.

**Table 1.** Spherical model parameters generated via genetic algorithm for first and second generation.

| Parameter | $t$ | $P_1$ | $P_2$ | $P_3$ | $P_4$ | $C_1$ | $C_2$ | $C_3$ | $C_4$ | $C_5$ | $C_6$ | $C_7$ | $C_8$ |
|---|---|---|---|---|---|---|---|---|---|---|---|---|---|
| $h$ | 1 | 33.5896 | 29.1142 | 31.1072 | 28.2664 | 24.8035 | 30.58 | 33.3301 | 35.1367 | 27.9462 | 28.6664 | 31.0478 | 32.0006 |
| $k$ | 1 | 44.2103 | 40.3246 | 38.2204 | 39.1703 | 37.6548 | 42.0848 | 43.1417 | 47.8128 | 36.1213 | 38.5753 | 38.8636 | 44.1221 |
| $l$ | 1 | 21.46 | 31.1081 | 35.0979 | 25.7211 | 20.0535 | 21.7505 | 30.3015 | 34.3909 | 23.018 | 30.1367 | 32.9515 | 37.372 |
| *fitness* | | 1.9963 | 0.5305 | 1.4442 | 0.9236 | 1.9747 | 1.1481 | 1.9987 | 4.5804 | 1.774 | 0.9567 | 0.9154 | 2.9193 |
| $h$ | 2 | 29.1142 | 30.58 | 28.2664 | 31.0478 | 24.1021 | 29.2203 | 30.1932 | 25.8899 | 24.7351 | 29.657 | 29.8396 | 28.2141 |
| $k$ | 2 | 40.3246 | 42.0848 | 39.1703 | 38.8636 | 40.0788 | 41.0045 | 41.4932 | 41.898 | 35.5124 | 38.9784 | 39.0833 | 38.8631 |
| $l$ | 2 | 31.1081 | 21.7505 | 25.7211 | 32.9515 | 20.5784 | 24.3133 | 27.9526 | 28.4532 | 21.9167 | 27.3088 | 32.5313 | 30.1781 |
| *fitness* | | 0.5305 | 1.1481 | 0.9236 | 0.9154 | 1.872 | 0.8474 | 0.7176 | 1.7872 | 2.4692 | 0.6377 | 0.6918 | 0.9849 |

### 2.2. Micro-Scale Cylindrical Surface Modeling via Genetic Algorithm

The micro-scale cylindrical surface modeling is performed by a genetic algorithm via surface coordinates, which are retrieved via micro laser line projection. To carry it out, a cylindrical surface model is constructed by means of surface coordinates, which are retrieved from a rectangular cylindrical surface. The surface coordinates are shown in Figure 3, where, the surface coordinates are represented by $(x_{i,j}, y_{i,j}, z_{i,j})$, and the subindices $(I, j)$ are defined in the $x$-axis and $y$-axis, respectively. Thus, the rectangular surface provides the surface points $(z_{0,0}, z_{1,0}, \ldots, z_{n,0}, z_{n,1}, \ldots, z_{n,m})$ in the $z$-axis, where, the sub-indices $(n, m)$ depict the number of surface points in the $x$-axis and $y$-axis, respectively. The origin coordinates are represented by $(x_{0,j}, k_0, l_0)$, the direction coordinates are depicted by $(x_{n,j}, k_n, l_n)$, and the ratio is defined by $r$. In this way, the cylindrical surface model is defined based on the circumference equation via ratio $r$ by means of the next expression

$$(y_{i,j} - k_i)^2 + (z_{i,j} - l_i)^2 = r^2 \tag{12}$$

For this equation, the coordinates $(k_i, l_i)$ are determined based on the surface geometry shown in Figure 3 by means of the next expressions

$$k_i = \frac{(k_n - k_{0,})}{(x_{n,j} - x_{0,j})}(x_{i,j} - x_{0,j}) + k_0 \tag{13}$$

$$l_i = \frac{(l_n - l_0)(k_i - k_0)}{(k_n - k_0)} + l_0 \tag{14}$$

These equations are computed via origin and direction coordinates. In this way, the cylindrical surface model is performed by determining the coordinates $(k_0, l_0)$, $(k_n, l_n)$ and the ratio $r$ via genetic algorithm.

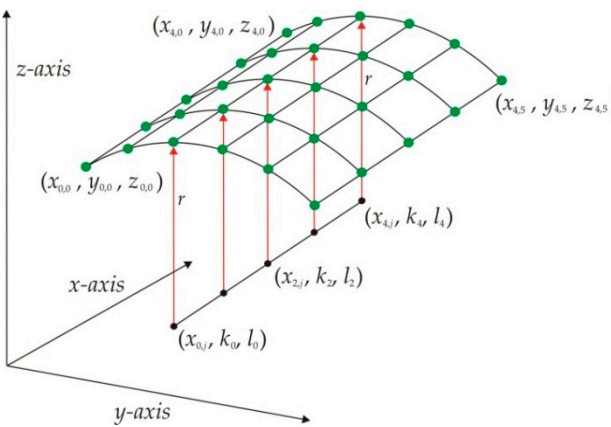

**Figure 3.** Rectangular cylindrical surface to perform cylindrical surface modeling.

To do so, the genetic algorithm determines the origin and direction coordinates via Equation (12), and surface coordinates $(x_{i,j}, y_{i,j}, z_{i,j})$. This procedure is carried out in five steps, which are described as follows:

The first step is to determine the initial population via surface data. To do so, the genetic algorithm computes the initial population of the parameters $(k_0, l_0, k_n, l_n)$ by means of the next expressions

$$k_0 = \frac{k_{i+\delta}(x_{n,j} - x_{0,j})(x_{i,j} - x_{0,j}) - k_i(x_{n,j} - x_{0,j})(x_{i+\delta,j} - x_{0,j})}{[(x_{i,j} - x_{n,j})(x_{i+\delta,j} - x_{0,j}) - (x_{i+\delta,j} - x_{n,j})(x_{i,j} - x_{0,j})]} \tag{15}$$

$$l_0 = \frac{(k_n - k_0)[l_{i+\delta}(k_i - k_0) - l_i(k_{i+\delta} - k_0)]}{(k_i - k_n)(k_{i+\delta} - k_0) - (k_{i+\delta} - k_n)(k_i - k_0)} \tag{16}$$

$$k_n = \frac{k_i(x_{n,j} - x_{0,j})(x_{i+\delta,j} - x_{n,j}) - k_{i+\delta}(x_{n,j} - x_{0,j})(x_{i,j} - x_{n,j})}{(x_{i,j} - x_{0,j})(x_{i+\delta,j} - x_{n,j}) - (x_{i+\delta,j} - x_{0,j})(x_{i,j} - x_{n,j})} \tag{17}$$

$$l_n = \frac{l_{i+\delta}(k_n - k_0)(k_n - k_i) - l_i(k_n - k_0)(k_n - k_{i+\delta})}{(k_{i+\delta} - k_0)(k_n - k_i) - (k_i - k_0)(k_n - k_{i+\delta})} \tag{18}$$

For these equations, the sub-index $\delta$ represents an increment in the $x$-axis, and the parameters $(k_i, l_i)$ are determined via surface data by means of the next expressions

$$k_i = \frac{\mathcal{Z}_1[\mathcal{Y}_4 + \mathcal{Z}_4] - \mathcal{Z}_3[\mathcal{Y}_2 + \mathcal{Z}_2]}{2[\mathcal{Y}_3\mathcal{Z}_1 - \mathcal{Y}_1\mathcal{Z}_3]} \tag{19}$$

$$l_i = \frac{\mathcal{Y}_1[\mathcal{Y}_4 + \mathcal{Z}_4] - \mathcal{Y}_3[\mathcal{Y}_2 + \mathcal{Z}_2]}{2[\mathcal{Z}_3\mathcal{Y}_1 - \mathcal{Z}_1\mathcal{Y}_3]} \tag{20}$$

where, $\mathcal{Z}_1 = (z_{i,j+\Delta}) - (z_{i,j})$, $\mathcal{Z}_2 = (z_{i,j+\Delta})^2 - (z_{i,j})^2$, $\mathcal{Z}_3 = (z_{i,j+\kappa}) - (z_{i,j})$, $\mathcal{Z}_4 = (z_{i,j+\kappa})^2 - (z_{i,j})^2$, $\mathcal{Y}_1 = (y_{i,j+\Delta}) - (y_{i,j})$, $\mathcal{Y}_2 = (y_{i,j+\Delta})^2 - (y_{i,j})^2$, $\mathcal{Y}_3 = (y_{i,j+\kappa}) - (y_{i,j})$, and $\mathcal{Y}_4 = (y_{i,j+\kappa})^2 - (y_{i,j})^2$. In this case, the sub-indices $\Delta$ and $\kappa$ represent an increment in the $y$-axis. By computing Equations (15)–(18), four parents $(P_{1,t}, P_{2,t}, P_{3,t}, P_{4,t})$ are determined for each parameter. The sub-index $t$ indicates the generation number. Thus, the parent $P_{1,1}$ is determined by computing Equations (15)–(18) via sub-indices ($i = 1, j = 1, \delta = n/4, \Delta = m/2, \kappa = 2\Delta$), the parent $P_{2,1}$ is determined via sub-indices ($i = n/4, j = 0, \delta = n/4, \Delta = m/2, \kappa = 2\Delta$), the parent $P_{3,1}$ is determined via sub-indices ($i = n/2, j = 0, \delta = n/4, \Delta = m/2, \kappa = 2\Delta$), and the parent $P_{4,1}$ is determined via sub-indices ($i = 3n/4, j = 0, \delta = n/4, \Delta = m/2, \kappa = 2\Delta$). In this way, four coordinates $(k_0, l_0, k_n, l_n)$ are obtained and they are defined as the initial population.

Additionally, the ratio $\check{r}$ is defined as the average distance between the coordinates $(k_i, l_i)$ and the surface coordinates $(x_{i,j}, y_{i,j}, z_{i,j})$. Then, the genetic algorithm determines the search space based on the maximum and minimum of each parameter. For the parameter $k_0$, the minimum and maximum are defined as $y_{0,0}$ and $y_{0,m}$, respectively. For $k_n$, the minimum and maximum are defined as $y_{n,0}$ and $y_{0,m}$, respectively. For $l_0$, the minimum and maximum are defined as $(l_0 + \check{r}/2)$ and $(l_0 - \check{r})$, respectively, where, $l_0$ is computed via Equation (16). For $l_n$, the minimum and maximum are defined as $(l_n + \check{r}/2)$ and $(l_n - \check{r})$, respectively, where, $l_n$ is computed via Equation (18). Thus, the initial population is completed.

The second step is to create the current children via crossover. To do so, two children are created inside parents and two children are created outside parents. Thus, the children $(C_{1+4^*q,t}, C_{2+4^*q,t}, C_{3+4^*q,t}, C_{4+4^*q,t})$ are created via parents $(P_{1+2^*q,t}, P_{2+2^*q,t})$ for $q = 0$ and $q = 1$. By replacing the parents $(P_{1,t}, P_{2,t})$ and $q = 0$ in Equations (7)–(10), the children $(C_{1,t}, C_{2,t}, C_{3,t}, C_{4,t})$ are determined. In addition, the parents $(P_{3,t}, P_{4,t})$ and $q = 1$ are replaced in Equations (7)–(10) to compute the children $(C_{5,t}, C_{6,t}, C_{7,t}, C_{8,t})$.

The third step is to evaluate the fitness. To do so, an objective function is constructed via the surface data. The objective function is defined by means of the next expression

$$\mathcal{FO} = \min\left\{ \frac{1}{nxm} \sum_{i=0}^{n} \sum_{j=0}^{m} \left[ \breve{r} - \sqrt{(y_{i,j} - k_i)^2 + (z_{i,j} - l_i)^2} \right]^2 \right\} \tag{21}$$

$$\breve{r} = \frac{1}{nxm} \sum_{i=0}^{n} \sum_{j=0}^{m} \left[ \sqrt{(y_{i,j} - k_i)^2 + (z_{i,j} - l_i)^2} \right]$$

For these equations, $k_i$ and $l_i$ are computed via Equations (13) and (14), respectively. Thus, the fitness of parents and children is computed.

The fourth step is to select the parents of the next generation. To do so, the best current parents and children are selected via fitness. Thus, the parents $P_{1,t+1}$ and $P_{3,t+1}$ are selected from the pairs $(P_{1,t}, P_{2,t})$ and $(P_{3,t}, P_{4,t})$, respectively. The parents $P_{2,t+1}$ and $P_{4,t+1}$ are selected from the children $(C_{1,t}, C_{2,t}, C_{3,t}, C_{4,t})$ and $(C_{5,t}, C_{6,t}, C_{7,t}, C_{8,t})$, respectively.

The fifth step is to perform the mutation to avoid trapping in a local minimum. To do so, the worst parent is selected by computing the fitness via Equation (21). Then, a new parent is randomly generated from the search space and the fitness is computed via Equation (19). If the new parent improves the fitness, the worst parent is replaced by the new parent, if not, the worst parent is not mutated.

Additionally, a parent is randomly selected to mutate one parameter, which is selected in random form. To carry it out, a new parameter is randomly generated from the search space. Then, the new parameter is replaced in the selected parent and the fitness is computed via Equation (21). If the parent fitness is improved, the parameter is replaced by the new coordinate, if not, the parameter is not mutated. Thus, the mutation of the current parents is completed and the parents of the $(t + 1)$ generation are determined. Then, the crossover is carried out to create the children of the $(t + 1)$ generation via Equations (7)–(10). Additionally, the fitness of these children is computed via Equation (19). Thus, the population of the $(t + 1)$ generation is completed. The procedure to compute the $(t + 1)$ generation is repeated until the parameters $(k_0, l_0, k_n, l_n)$ that minimize the objective function in Equation (21) are obtained. From these parameters, the cylindrical surface model is defined by the expression $z_{i,j} = l_i + [\check{r}^2 - (y_{i,j} - k_i)^2]^{1/2}$. Thus, the cylindrical surface model is determined by the metaheuristic algorithm.

### 2.3. Micro-Scale Surface Contouring via Micro-Laser Line Projection

The micro-scale surface is retrieved by the microscope vision system shown in Figure 4a. This setup consists of an optical microscope, a laser diode, a CCD camera, a slider device, and a computer. The surface plane is located on the $x$-axis and $y$-axis, and the surface topography is parallel to the $z$-axis. The lateral configuration of the microscope vision system in the $x$-axis is depicted by the geometry shown in Figure 4b, where a 42 μm laser

line is projected perpendicularly on the target surface and the optical microscope is aligned at an angle. The symbol $\theta$ indicates the angle between the laser line and the optical axis. The distance between the objective lens and the surface is represented by $d$. The objective focal length is depicted by $f_1$ and the objective focus is represented by $F_1$. The distance between the intermediate image plane and the ocular lens is depicted by $L$. The ocular focal length is indicated by $f_2$ and the ocular focus is represented by $F_2$. The lateral configuration of the microscope vision system in the $y$-axis is described by means of the geometry shown in Figure 4c. The laser line coordinates in the CCD array are denoted by $(x_{i,j}, y_{i,j})$ in the $x$-axis and $y$-axis, respectively.

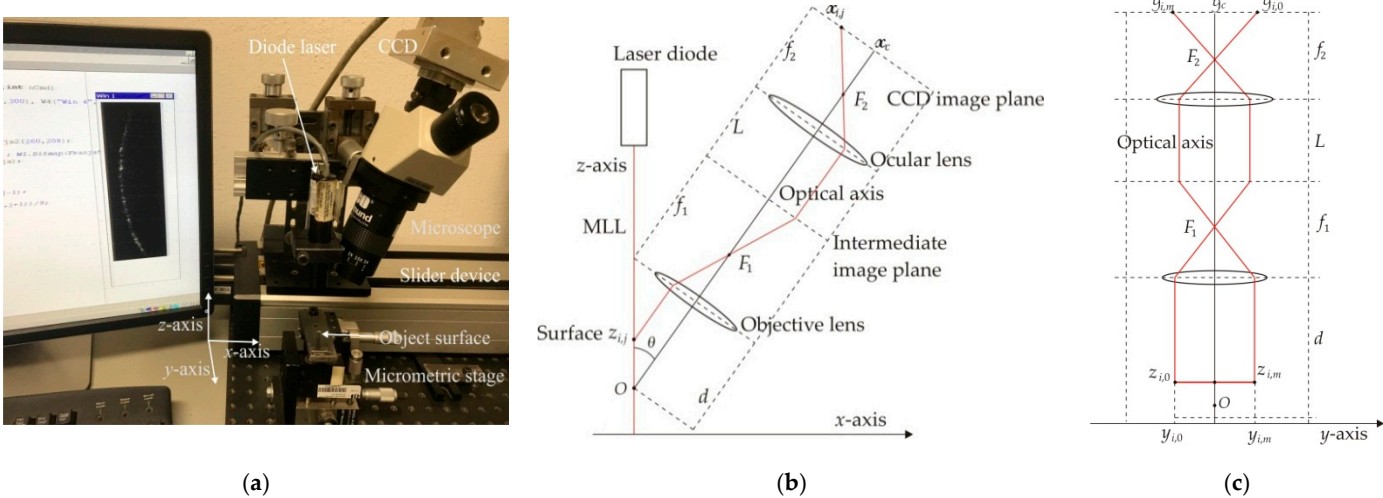

(**a**) 　　　　　　　　　　　(**b**) 　　　　　　　　　　　(**c**)

**Figure 4.** (**a**) Optical microscope vision system to perform micro-scale spherical and cylindrical surface modeling. (**b**) Lateral configuration of the microscope vision system in $x$-axis. (**c**) Lateral configuration of the microscope vision system in $y$-axis.

The surface coordinates $y_{i,j}$ and $z_{i,j}$ are defined via geometry shown in Figure 4b,c by means of the expressions

$$z_{i,j} = \frac{\eta(x_c - x_{i,j})F_1 F_2}{(f_1 - F_1)(f_2 - F_2)\sin\theta} + O \tag{22}$$

$$y_{i,j} = \eta y_c - \frac{\eta(y_c - y_{i,j})F_1 F_2}{(f_1 - F_1)(f_2 - F_2)} \tag{23}$$

The surface coordinate $x_{i,j}$ is defined by the position where the laser line is projected in the $x$-axis. This coordinate is provided by the slider device. In this way, the surface topography is computed based on the vision parameters $(x_c, y_c, \eta, \theta, f_1, F_1, f_2, F_2)$. These parameters are determined via genetic algorithm by means of Equations (22) and (23), and surface data. The genetic algorithm determines the vision parameters in five steps, which are described as follows.

The first step is to determine the initial population. To do so, the search space is deduced via the maximum and minimum of each parameter. The maximum and minimum of the parameters $(x_c, y_c, \eta)$ are deduced via image size. The maximum and minimum of the parameters $(\theta, f_1, F_1, f_2, F_2)$ are defined via microscope geometry. The minimum $F_2$ is defined by a distance more than half of the ocular lens diameter, and the maximum $F_2$ is two times the minimum $F_2$. The minimum $f_2$ is defined as the minimum $F_2$ and the maximum $f_2$ is four times the minimum $F_2$. In the same way, the minimum $F_1$ is defined by a distance more than half of the objective lens diameter, and the maximum $F_1$ is two times the minimum $F_2$. The minimum $f_1$ is determined as the minimum $F_1$ and the maximum $f_1$ is four times the minimum $F_1$. The minimum $\theta$ is defined as an angle more than 10° and

the maximum $\theta$ is defined as an angle less than $45°$. Thus, the search space is determined. Then, four parents ($P_{1,t}$, $P_{2,t}$, $P_{3,t}$, $P_{4,t}$) are created in random form from the search space. In this way, four values of the vision parameters ($x_c$, $y_c$, $\eta$, $\theta$, $f_1$, $F_1$, $f_2$, $F_2$) are obtained and they are defined as the initial population.

The second step is to create the current children via crossover. To do so, the children ($C_{1+4*q,t}$, $C_{2+4*q,t}$, $C_{3+4*q,t}$, $C_{4+4*q,t}$) are created via parents ($P_{1+2*q,t}$, $P_{2+2*q,t}$) for $q = 0$ and $q = 1$. By replacing the parents ($P_{1,t}$, $P_{2,t}$) and $q = 0$ in Equations (7)–(10), the children ($C_{1,t}$, $C_{2,t}$, $C_{3,t}$, $C_{4,t}$) are determined. In addition, the parents ($P_{3,t}$, $P_{4,t}$) and $q = 1$ are replaced in Equations (7)–(10) to compute the children ($C_{5,t}$, $C_{6,t}$, $C_{7,t}$, $C_{8,t}$).

The third step is to evaluate the fitness. To do so, an objective function is constructed based on the known surface data and the vision parameters. The objective function is defined by means of the next expressions

$$FO_1 = \min\left\{ \frac{1}{mxn} \sum_{i=0}^{n} \sum_{j=0}^{m} \left[ (z_{i,j} - z_{i,m}) - \frac{\eta(x_c - x_{i,j})F_1 F_2}{(f_1 - F_1)(f_2 - F_2)sin\theta} + \frac{\eta(x_c - x_{i,m})F_1 F_2}{(f_1 - F_1)(f_2 - F_2)sin\theta} \right]^2 \right\} \quad (24)$$

$$FO_2 = \min\left\{ \frac{1}{mxn} \sum_{i=0}^{n} \sum_{j=0}^{m} \left[ (y_{i,j} - y_{i,m}) + \frac{\eta(y_c - y_{i,j})F_1 F_2}{(f_1 - F_1)(f_2 - F_2)} - \frac{\eta(y_c - y_{i,m})F_1 F_2}{(f_1 - F_1)(f_2 - F_2)} \right]^2 \right\} \quad (25)$$

From these equations, the fitness is computed by the expression $FO = (FO_1 + FO_2)/2$. In this case, the surface topography ($z_{i,j} - z_{i,m}$) and the surface width ($y_{i,j} - y_{i,1}$) are known.

The fourth step is to select the parents of the next generation. To do so, the best current parents and children are selected via fitness. Thus, the parents $P_{1,t+1}$ and $P_{3,t+1}$ are selected from the pairs ($P_{1,t}$, $P_{2,t}$) and ($P_{3,t}$, $P_{4,t}$), respectively. Then, the parents $P_{2,t+1}$ and $P_{4,t+1}$ are selected from the children ($C_{1,t}$, $C_{2,t}$, $C_{3,t}$, $C_{4,t}$) and ($C_{5,t}$, $C_{6,t}$, $C_{7,t}$, $C_{8,t}$), respectively.

The fifth step is to perform the mutation to avoid trapping in a local minimum. To do so, the worst parent is selected by computing the fitness via Equations (24) and (25). Then, a new parent is randomly generated from the search space and the fitness is computed. If the new parent improves the fitness, the worst parent is replaced by the new parent, if not, the worst parent is not mutated. Additionally, a parent is randomly selected to mutate one vision parameter, which is selected in random form. To carry it out, a new vision parameter is randomly generated from the search space. Then, the new vision parameter is replaced in the selected parent and the fitness is computed via Equations (24) and (25). If the parent fitness is improved, the vision parameter is replaced by the new parameter, if not, the vision parameter is not mutated. Thus, the mutation of the current parents is completed and the parents of the ($t + 1$) generation are determined. Then, the crossover is carried out to create the children of the ($t + 1$) generation via Equations (7)–(10). In addition, the fitness of these children is computed via Equations (24) and (25). Thus, the population of the ($t + 1$) generation is completed. The procedure to compute the ($t + 1$) generation is repeated until the vision parameters ($x_c$, $y_c$, $\eta$, $\theta$, $f_1$, $F_1$, $f_2$, $F_2$) that minimize the objective function Equations (24) and (25) are obtained. Additionally, the distance from zero to the point $O$ is determined by the expression $z_{0,j} = \eta(x_{0,j} - x_c) F_1 F_2 / (f_1 - F_1)(f_2 - F_2)sin\theta$.

The micro laser line coordinates ($x_{i,j}$, $y_{i,j}$) are determined via pixel position and intensity. The coordinate $x_{i,j}$ is computed through the maximum intensity in each row of the image [47]. To do so, the pixel position and intensity are fitted to a Bezier curve by means of the expressions

$$x(u) = \sum_{i=0}^{N} C_i(1 - u)^{N-i} u^i x_{i,j}, \; C_i = C_{i-1}(N + 1 - i)/i, \; C_0 = 1, \; 0 \leq u \leq 1 \quad (26)$$

$$I(u) = \sum_{i=0}^{N} C_i(1 - u)^{N-i} u^i I_{i,j}, \; C_i = C_{i-1}(N + 1 - i)/i, \; C_0 = 1, \; 0 \leq u \leq 1 \quad (27)$$

For Equation (26), $x_{i,j}$ is the pixel coordinate of the micro laser line in the $x$-axis and $N$ is the pixels number of the micro laser line. For Equation (27), $I_{i,j}$ is the pixel intensity. In this case, the sub-indices $(i, j)$ indicate the pixel number of the micro laser line in the $x$-axis and $y$-axis, respectively. Thus, the pixel position $x_{i,j}$ is replaced in Equation (26) and the intensity $I_{i,j}$ is substituted in Equation (27) to obtain a curve $\{x(u), I(u)\}$. In this case, a concave curve is obtained in the interval $0 \leq u \leq 1$. Therefore, the second derivative $I''(u)$ is positive, and the maximum intensity is determined by means of the derivative $I'(u) = 0$. In this way, the Bisection method is employed to calculate the value $u$, where $I'(u) = 0$. Then, the value $u$ is substituted in Equation (26) to compute $x(u)$, which is defined as the micro laser line position $x_{i,j} = x(u)$ in the $x$-axis. The micro laser line position $y_{i,j}$ is determined via image row number. The micro laser line edges $y_{i,0}$ and $y_{i,m}$ are calculated by means of the first derivative in the $y$-axis. Thus, the micro laser line coordinates $(x_{i,j}, y_{i,j})$ are obtained to compute the micro-scale surface topography. To do so, the microscope vision system scans the surface to perform the micro-scale surface modeling. During the scanning, the CCD camera captures the micro laser line to compute the line coordinates $(x_{i,j}, y_{i,j})$ via Equations (26) and (27). Then, the coordinates $(x_{i,j}, y_{i,j})$ are substituted in Equations (22) and (23) to compute the surface coordinates $(z_{i,j}, y_{i,j})$. The surface coordinate in the $x$-axis corresponds to the position where the micro laser line is projected. This coordinate is provided by the slider device. Thus, the micro-scale topography is retrieved.

The radial distortion is determined via the micro laser line position. The micro laser line position $(x_{i,j}, y_{i,j})$ is computed via Equations (26) and (27). Thus, the radial distortion is deduced by means of the line position $x_{i,j} = x_{i,j} + \delta x_i$ and $y_{i,j} = y_{i,j} + \delta y_j$, where $(x_{i,j}, y_{i,j})$ are the distorted coordinates and $(\delta x_i, \delta y_j)$ are the distortion. In addition, a distorted line shifting is determined by the expression $S_{i,j} = x_{1,j} - x_{i,j}$, and the undistorted line shifting is defined by the expression $s_{i,j} = (x_{1,j} + \delta x_1) - (x_{i,j} + \delta x_i)$. From these expressions, the distortion in the $x$-axis is determined by means of the expression $\delta x_i = (x_{1,j} - x_{i,j}) - s_{i,j} + \delta x_1 = S_{i,j} - s_{i,j} + \delta x_1$. By placing the laser line near to the image center, the first line shifting is obtained without distortion, where, $\delta x_1 = 0$, and $s_{1,j} = S_{1,j}$. Thus, the undistorted shifting $s_{i,j}$ is computed through the first shifting $S_{1,j}$ by means of the expression $s_{i,j} = i \times S_{1,j}$. Therefore, the distortion in the $x$-axis is calculated by the expression $\delta x_i = (x_{1,j} - x_{i,j}) - i \times S_{1,j}$. The same procedure is performed to determine the distortion in the $y$-axis via expressions $(y_{i,1} - y_{i,j}) = (y_{i,1} + \delta y_1) - (y_{i,j} + \delta y_j)$ and $T_{i,j} = (y_{i,1} - y_{i,j})$. From this relationship, the distortion in the $y$-axis is calculated by the expression $\delta y_j = (y_{i,1} - y_{i,j}) - j^* T_{i,1}$.

### 3. Results of Micro-Scale Spherical and Cylindrical Surface Modeling

The micro-scale spherical and cylindrical surface modeling is carried out by the microscope vision system shown in Figure 4a. In this way, the micro-scale spherical surface modeling is performed for the plastic surface shown in Figure 5a. The scale of this figure is represented in millimeters in the $x$-axis. The micro laser line projected on the plastic surface is shown in Figure 5b. Thus, the plastic surface is scanned in the $x$-axis via the micro laser line to compute the coordinates $(x_{i,j}, y_{i,j})$ by means of Equations (26) and (27). Then, the coordinates $(x_{i,j}, y_{i,j})$ are substituted in Equations (23) and (23) to compute the micro-scale surface coordinates $(z_{i,j}, y_{i,j})$. The surface coordinate $x_{i,j}$ is provided by the slider device. Thus, one hundred and ninety four images are processed to retrieve the micro-scale spherical surface shown in Figure 6a. In this figure, the $x$-axis, $y$-axis, and $z$-axis are indicated in microns. The accuracy of the surface shown in Figure 6a is determined via relative error [48], which is computed by employing a reference contact method. Thus, the relative error is computed by the next expression

$$Error\% = \frac{100}{n \cdot m} \sum_{i=0}^{n} \sum_{j=0}^{m} \frac{|z_{i,j} - H_{i,j}|}{H_{i,j}} \tag{28}$$

For this equation, $z_{i,j}$ is the micro-scale surface computed via Equation (22), $H_{i,j}$ is the surface reference, and $n \cdot m$ is the data number. By computing Equation (28), the relative error for the plastic surface shown in Figure 6a is 0.8216%. From the surface shown in Figure 6a, a spherical surface model is constructed via genetic algorithm, which determines the origin coordinates $(h, k, l)$ as described in Section 2.1.

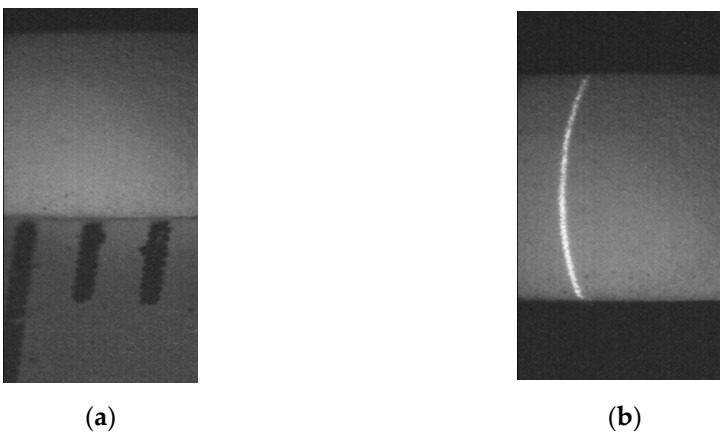

(**a**)　　　　　　　　　　　　　(**b**)

**Figure 5.** (**a**) Rectangular spherical surface to perform spherical surface modeling. (**b**) Micro laser line projected on the spherical surface to retrieve the surface topography.

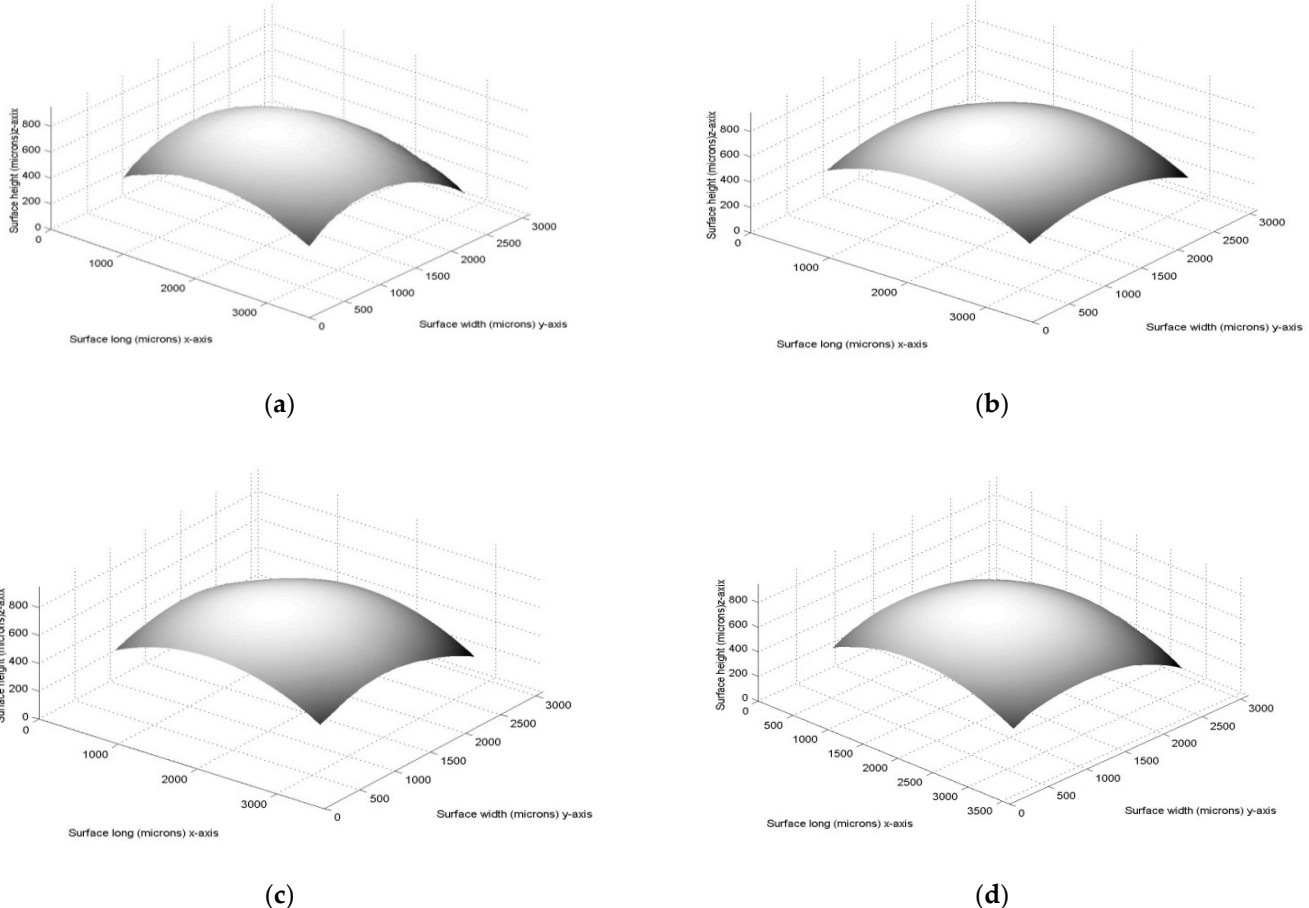

(**a**)　　　　　　　　　　　　　(**b**)

(**c**)　　　　　　　　　　　　　(**d**)

**Figure 6.** (**a**) Spherical surface retrieved via micro laser line scanning. (**b**) Spherical surface generated via spherical surface model Equation (1). (**c**) Surface generated via spherical surface model Equation (1) and the surface data over the surface model. (**d**) Surface generated via spherical surface model Equation (1) and surface data under the model.

In this way, the genetic algorithm performs the first step to determine the initial population. To carry it out, the search space is defined based on the maximum and minimum of each parameter. For the spherical surface model, the minimum and maximum of the parameter $h$ are defined as $x_{0,0}$ and $x_{n,m}$, respectively. The minimum and maximum of the parameter $k$ are defined as $y_{0,0}$ and $y_{n,m}$, respectively. The minimum and maximum of the parameter $l$ are defined as $(l + \check{R}/2)$ and $(l - \check{R})$, respectively. Then, the initial population is determined. Typically, the metaheuristic algorithms determine the initial population from the search space in random form. However, a huge iterations number is employed to achieve the convergence. Therefore, the algorithms employ the best candidates to improve the solution quality and to speed out the convergence. The initial population plays a significant role in the genetic algorithm to provide higher qualities and better convergence [49]. Generally, the genetic algorithms generate the initial population employing the best candidates. These candidates can be obtained via random methods and by sampling the model equations [50,51]. Based on these statements, the proposed genetic algorithm performs a sampling of the model to obtain the best candidates. The sampling is carried out by solving Equations (2) and (3) to obtain the best candidates to perform the optimization. To do so, the spherical surface model is sampled at the surface coordinates $(x_{i,j}, y_{i,j}, z_{i,j})$, $(x_{i+\delta,j}, y_{i+\delta,j}, z_{i+\delta,j})$, $(x_{i,j+\Delta}, y_{i,j+\Delta}, z_{i,j+\Delta})$, and $(x_{i+\delta,j+\Delta}, y_{i+\delta,j+\Delta}, z_{i+\delta,j+\Delta})$ to obtain a good initial population $(h, k, l)$. In this way, Equations (4)–(6) are computed by employing $m = 180$, $n = 194$, $\Delta = 90$, and $\delta = 97$. Thus, the parent $P_{1,1}$ is determined via surface data whose sub-indices are ($i = 0$, $j = 0$, $\delta = 97$, $\Delta = 90$), the parent $P_{2,1}$ is determined via ($i = 0$, $j = 90$, $\delta = 97$, $\Delta = 90$), the parent $P_{3,1}$ is determined via sub-indices ($i = 97$, $j = 0$, $\delta = 97$, sub-indices $\Delta = 90$), and the parent $P_{4,1}$ is determined via sub-indices ($i = 97$, $j = 90$, $\delta = 97$, $\Delta = 90$).

By employing these surface data, four origin coordinates $(h, k, l)$ are computed via Equations (4)–(6). From this procedure, the parent $P_{1,1}$ is represented by ($h = 1218.8$ μm, $k = 1173.2$ μm, $l = -3135.7$ μm), the parent $P_{2,1}$ is represented by ($h = 1672.1$ μm, $k = 1389.1$ μm, $l = -3813.6$ μm), the parent $P_{3,1}$ is represented by ($h = 1302.1$ μm, $k = 1276.6$ μm, $l = -3941.3$ μm), and the parent $P_{4,1}$ is determined by ($h = 1762.91$ μm, $k = 1516.8$ μm, $l = -3584.3$ μm). These parameters represent the initial population, which provides good fitness to speed out the convergence. In addition, the ratio $\check{R}$ is computed via Equation (11) by employing the origin coordinates $(h, k, l)$ and the surface points $(x_{i,j}, y_{i,j}, z_{i,j})$.

Then, the genetic algorithm performs the second step via crossover to compute the children by substituting $q = 0$ and $q = 1$ in Equations (7)–(10). The probability of crossover is determined via fitness [52]. When the average fitness of the parents is improved, the crossover is carried out. This procedure leads to preventing the loss of candidates to achieve the convergence. Thus, the probability of crossover is in the interval from 0.0 to 0.5. This result is obtained by computing the same experiment several times. Then, the algorithm performs the third step to evaluate the fitness by substituting the coordinates $(h, k, l)$ of parents and children in Equation (11). Then, the algorithm performs the fourth step to select the parents of the next generation via fitness. In this way, the parents $P_{1,t+1}$ and $P_{3,t+1}$ are selected from the pairs $(P_{1,t}, P_{2,t})$ and $(P_{3,t}, P_{4,t})$, respectively. Furthermore, the parents $P_{2,t+1}$ and $P_{4,t+1}$ are selected from the children $(C_{1,t}, C_{2,t}, C_{3,t}, C_{4,t})$ and $(C_{5,t}, C_{6,t}, C_{7,t}, C_{8,t})$, respectively.

Then, the algorithm performs the fifth step to mutate the worst parent, which is determined via fitness. In addition, a parent is randomly selected to mutate one coordinate, which is selected in random form. The mutation probability is determined via fitness. For the mutation of the parent, if the new parent improves the fitness, the worst parent is mutated, if not, the worst parent is not mutated. For the parameter mutation, if the new parameter improves the parent fitness, the parameter is mutated. Thus, the mutation probability crossover is in the interval from 0.0 to 0.62. This result is obtained by computing the same experiment several times. Then, the algorithm performs the second step to create the children of the $(t + 1)$ generation via Equations (7)–(10). Additionally, the fitness of these children is computed via Equation (11). Thus, the population of the $(t + 1)$ generation is completed. Then, the procedure to compute the $(t + 1)$ generation is repeated until the op-

timal parameters $(h, k, l)$, which minimize the objective function Equation (11) are obtained. The number of generations is deduced from the iterations number to obtain the optimized parameters $(h, k, l)$. Thus, the number of generations that obtain the surface model is 136. The optimized origin coordinates are $h$ = 1688.3 µm, $k$ = 1409.4 µm, $l$ = −3685.6 µm, and $R$ = 4390.9 µm. These coordinates are substituted in Equation (1) to obtain the spherical surface model $z_{i,j} = -3685.6 + [4390.9^2 - (x_{i,j}-1688.3)^2 - (y_{i,j} - 1409.4)^2]^{1/2}$, which produces the spherical surface shown in Figure 6b. The relative error produced by the spherical model is determined by the next expression

$$error\% = \frac{100}{n \cdot m} \sum_{i=0}^{n} \sum_{j=0}^{m} \frac{\left| M_{i,j} - z_{i,j} \right|}{z_{i,j}} \tag{29}$$

For this equation, $M_{i,j}$ is the micro-scale surface computed via Equation (1), $z_{i,j}$ is the surface retrieved via micro laser line projection, and $n \cdot m$ is the data number. Thus, Equation (29) determines the optimality gap of the genetic algorithm [53]. By computing Equation (29), the relative error between the spherical surface model and the plastic surface shown in Figure 6b is 2.8631%. Additionally, Figure 6c shows the surface generated via the spherical surface model and the surface data over the surface model. Figure 6d shows the surface generated via the spherical surface model and the surface data under the surface model. The fitness variation with respect to the generation number is depicted by the graphic shown in Figure 7, where, the variation decreases when the generation number increases.

Thus, the optimality gap of the algorithm for the spherical surface model is 2.8631%. The solution quality is deduced based on the running time to obtain the optimal solution [54]. The running time is defined by the generation number and the computer speed. Thus, the 136 generations are computed in 22.87 s by employing a PC to 1.8 GHz. The fitness variation in each generation is shown in Figure 7. In this case, the variation decreases when the generation number increases. Thus, the spherical surface modeling is performed by the metaheuristic algorithm.

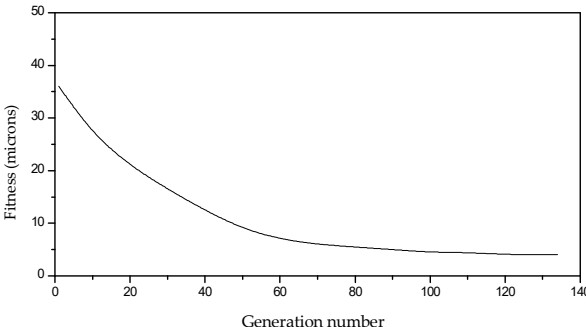

**Figure 7.** Fitness variation based on the number of generations.

The micro-scale cylindrical surface modeling is performed for the metallic surface shown in Figure 8a. The scale of this figure is represented in millimeters in the $x$-axis. The micro laser line projected on the metallic surface is shown in Figure 8b. Thus, the metallic surface is scanned in the $x$-axis via the micro laser line to compute the coordinates $(x_{i,j}, y_{i,j})$ by means of Equations (26) and (27). Then, the coordinates $(x_{i,j}, y_{i,j})$ are substituted in Equations (22) and (23) to compute the micro-scale surface $(z_{i,j}, y_{i,j})$. The surface coordinate $x_{i,j}$ is provided by the slider device. In this way, one hundred and eighty images are processed to retrieve the micro-scale cylindrical surface shown in Figure 9a. In this figure, the $x$-axis, $y$-axis, and $z$-axis are indicated in microns. The accuracy of this surface is determined via relative error in Equation (28). Where $z_{i,j}$ is the micro-scale surface computed via Equation (22), $H_{i,j}$ is the surface reference, and $n \cdot m$ is the data number. By computing Equation (28), the relative error for the metallic surface shown in Figure 9a is 0.8783%.

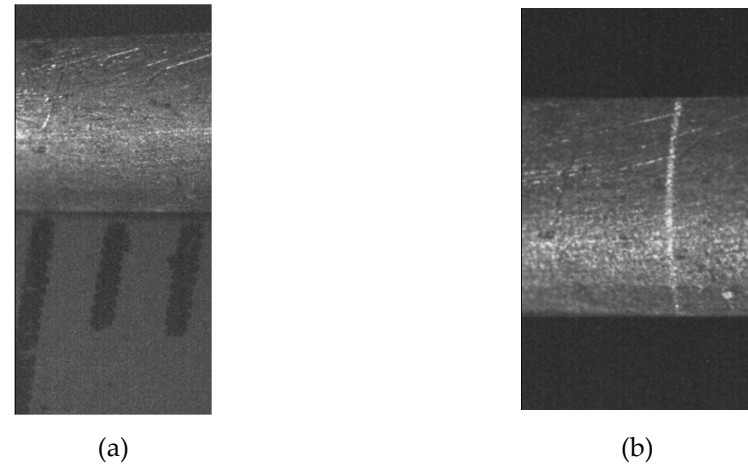

(a)                                                                (b)

**Figure 8.** (**a**) Rectangular cylindrical surface to perform cylindrical surface modeling. (**b**) Micro laser line projected on the cylindrical surface to retrieve the surface topography.

(**a**)                                                               (**b**)

(**c**)                                                               (**d**)

**Figure 9.** (**a**) Rectangular cylindrical surface to perform cylindrical surface modeling. (**b**) Micro laser line projected on the cylindrical surface to retrieve the surface topography. (**c**) Surface generated via cylindrical surface model and the surface data over the surface model. (**d**) Surface generated via cylindrical surface model and the surface data under the surface model.

From the surface shown in Figure 9a, a cylindrical surface model is performed by the genetic algorithm via surface coordinates $(x_{i,j}, y_{i,j}, z_{i,j})$. To carry it out, the genetic algorithm determines the origin and direction coordinates $(k_0, l_0, k_n, l_n)$ as described in Section 2.2. In this way, the genetic algorithm performs the first step to determine the initial population by employing the surface coordinates $(x_{i,j}, y_{i,j}, z_{i,j})$. Thus, a sampling of the model is carried out to obtain the best candidates. To do so, Equations (15)–(20) are computed by employing $m = 160$ and $n = 180$. Thus, the parent $P_{1,1}$ is computed via surface whose sub-indices are $(i = 0, j = 0, \delta = 45, \Delta = 80, \kappa = 160)$, the parent $P_{2,1}$ is determined via sub-indices $(i = 45, j = 0, \delta = 45, \Delta = 80, \kappa = 160)$, the parent $P_{3,1}$ is computed via sub-indices $(i = 90, j = 0, \delta = 45, \Delta = 80, \kappa = 160)$, and $P_{4,1}$ is determined via sub-indices $(i = 135, j = 0, \delta = 45, \Delta = 80, \kappa = 160)$. In this way, four coordinates $(k_0, l_0, k_n, l_n)$ are obtained and they are defined as the first parents $(P_{1,t}, P_{2,t}, P_{3,t}, P_{4,t})$. Additionally, the ratio $\check{r}$ is defined as the average distance between the coordinates $(k_i, l_i)$ and the surface coordinates $(x_{i,j}, y_{i,j}, z_{i,j})$. From this procedure, the parent $P_{1,1}$ is represented by the parameters $(k_0 = 1188.2\ \mu m, l_0 = -4496.4\ \mu m, k_n = 1236.5\ \mu m, l_n = -4328.2\ \mu m)$, the parent $P_{2,1}$ is defined by the parameters $(k_0 = 1291.5\ \mu m, l_0 = -4122.6\ \mu m, k_n = 1315.4\ \mu m\ l_n = -4110.5\ \mu m)$, the parent $P_{3,1}$ is represented by the parameters $(k_0 = 1346.2\ \mu m, l_0 = -4259.3\ \mu m, k_n = 1352.4\ \mu m, l_n = -4235.1\ \mu m)$, and the parent $P_{4,1}$ is determined by the parameters $(k_0 = 1401.2\ \mu m, l_0 = -4432.8\ \mu m, k_n = 1408.3\ \mu m, l_n = -4425.7\ \mu m)$. Furthermore, the ratio $\check{r}$ is computed via Equation (19) by employing the coordinates $(k_i, l_i)$ and the surface coordinates $(x_{i,j}, y_{i,j}, z_{i,j})$. Additionally, the minimum and maximum of the coordinate $k_0$ are defined as $y_{0,0}$ and $y_{0,m}$, respectively. The minimum and maximum of the coordinate $k_n$, are defined as $y_{n,0}$ and $y_{0,m}$, respectively. The minimum and maximum of the coordinate $l_0$ are defined as $(l_0 + \check{r}/2)$ and $(l_0 - \check{r})$, respectively, where, $l_0$ is computed via Equation (16). The minimum and maximum of the coordinate $l_n$ are defined as $(l_n + \check{r}/2)$ and $(l_n - \check{r})$, respectively, where, $l_n$ is computed via Equation (18). Then, the genetic algorithm performs the second step to compute the children by substituting $q = 0$ and $q = 1$ in Equations (7)–(10). The probability of crossover is determined via fitness. When the average fitness of the parents is improved, the crossover is carried out. Thus, the probability of crossover is in the interval from 0.0 to 0.48. This result is obtained by computing the same experiment several times. Then, the algorithm performs the third step to evaluate the fitness. In this procedure, Equation (21) is computed by means of the coordinates $(k_0, l_0, k_n, l_n)$ and the coordinates $(k_i, l_i)$, which are determined via Equation (13) and Equation (14). Then, the algorithm performs the fourth step to select the parents of the next generation. In this way, the parents $P_{1,t+1}$ and $P_{3,t+1}$ are selected from the pairs $(P_{1,t}, P_{2,t})$ and $(P_{3,t}, P_{4,t})$, respectively. Additionally, the parents $P_{2,t+1}$ and $P_{4,t+1}$ are selected from the children $(C_{1,t}, C_{2,t}, C_{3,t}, C_{4,t})$ and $(C_{5,t}, C_{6,t}, C_{7,t}, C_{8,t})$, respectively. Then, the algorithm performs the fifth step to mutate the worst parent, which is determined via fitness Equation (21). In addition, a parent is randomly selected to mutate one coordinate, which is selected in random form. The mutation probability is determined via fitness. If the parent mutation improves the fitness, the mutation is carried out. If not, the worst parent is not mutated. For the parameter mutation, if the new parameter improves the parent fitness, the parameter is mutated. Thus, the mutation probability crossover is in the interval from 0.0 to 0.65. This result is obtained by computing the same experiment several times. Then, the algorithm performs the second step to create the children of the $(t + 1)$ generation via Equations (7)–(10). Furthermore, the fitness of these children is computed via Equation (21). Thus, the population of the $(t + 1)$ generation is completed. The number of generations is deduced from the number of iterations to obtain the optimized parameters $(h, k, l)$. Thus, the number of generations to obtain the surface model is 112. Then, the procedure to compute the $(t + 1)$ generation is repeated until the optimal parameters $(k_0, l_0, k_n, l_n)$, which minimize the objective function Equation (19), are obtained. The optimized origin and direction coordinates are $k_0 = 1527.45\ \mu m, l_0 = -4633.36\ \mu m, k_n = 1528.51\ \mu m, l_n = -4622.34\ \mu m$, and $r = \check{r} = 4914.02\ \mu m$. These coordinates are substituted in Equation (12) to obtain the cylindrical surface model $z_{i,j} = l_i + [(4914.02)^2 - (y_{i,j} - k_i)^2]^{1/2}$, which produces the spherical surface shown in Figure 9b.

The error between the cylindrical surface model and the surface data shown in Figure 9a is computed via Equation (29), where, $M_{i,j}$ is the micro-scale surface computed via the cylindrical surface model in Equation (12), $z_{i,j}$ is the surface computed via micro laser line projection, and $n \cdot m$ is the data number. Thus, Equation (29) determines the optimality gap of the genetic algorithm. By computing Equation (29), the relative error between the cylindrical surface model and the surface data shown in Figure 9a is 3.765%.

Thus, the optimality gap of the algorithm for the cylindrical surface model is 3.765%. The solution quality is deduced based on the running time to obtain the optimal solution. The running time is defined by the generation number and the computer speed. Thus, the 112 generations are computed in 22.18 s by employing a PC to 1.8 GHz. Additionally, Figure 9c shows the surface generated via cylindrical surface model and the surface data over the surface model. Figure 9d shows the surface generated via cylindrical surface model and the surface data under the surface model. Thus, the cylindrical surface modeling has been performed by the metaheuristic algorithm.

## 4. Discussion

The capability of the micro-scale spherical and cylindrical surface modeling is established based on the model fitting accuracy [55,56]. Therefore, the contribution of the proposed technique is determined based on the model fitting to the surface. This statement includes the model fitting to the surface, and the algorithm efficiency. The model fitting to the surface and the algorithm efficiency are achieved in a good manner. The model fitting is determined via quality gap, which is computed via Equation (29). Thus, the proposed technique fits the spherical and cylindrical surface model to the surface data with a relative error smaller than 3.8%. The algorithm efficiency is determined via algorithm structure and solution quality, which is determined via running time. The algorithm structure provides an efficient procedure to optimize the surface model parameters. It is because the metaheuristic algorithm determines the initial population by the best candidates, which are sampled by solving the spherical and cylindrical surface models. This procedure leads to providing the initial population near the optimal parameters. Thus, the algorithm provides a low error since the first generation. This leads to reduced iterations to obtain the optimal parameters. Therefore, the running time is achieved in less than 140 iterations. Thus, the proposed micro-scale surface modeling improves the model fitting of the traditional optical microscope imaging systems. It is because the traditional optical microscope imaging systems perform micro-scale spherical and cylindrical surface modeling with a relative error over 5.0% [57,58], where, the spherical and cylindrical surface modeling is performed via polynomial regression by means of the least squared method [59,60]. In this way, a spherical model via least squared is generated for the plastic surface shown in Figure 6a and for the metallic shown in Figure 9a. However, the fitting accuracy is a relative error over 8%. This means that surface modeling via the metaheuristic algorithm improves the model fitting of the optical microscope imaging systems. Additionally, the surface data accuracy has an influence on the surface model fitting. In this matter, the optical microscope imaging systems perform the spherical and cylindrical surface modeling by means of surface, which is determined via gray-level [61,62]. However, the gray-level profile does not reproduce the surface profile under the test. Therefore, the gray-level does not help to improve the model fitting. Instead, the micro-scale contouring via micro laser line reproduces the surface profile with high accuracy. It is because the micro laser line reflection depicts the object surface profile. Additionally, the artificial intelligence algorithms have been employed by the contact methods to perform the spherical and cylindrical surface modeling [63,64]. These algorithms optimize the mathematical parameters of the spherical and cylindrical models by employing the traditional search structure [65,66]. In this matter, the proposed metaheuristic algorithm performs an analytic process to determine the search space, which provides initial values near to the optimal model parameters. This procedure is carried out via Equations (3)–(5) for the spherical surface modeling and Equations (15)–(18) for the cylindrical surface modeling. Thus, the metaheuristic algorithm determines the initial

parameters by solving the surface model via surface geometry data. These initial parameters are the best candidates for the first solution of the surface model parameters. In this way, the search procedure begins on a path, which provides the optimal parameters. It is because the search procedure begins near to the optimal parameters. This leads to improving the metaheuristic algorithm efficiency by reducing iterations to find the best model parameters. It is elucidated by the low error in the first generation. Thus, the metaheuristic algorithm determines the optimal parameters from the surface data. The traditional algorithms do not provide the initial parameters near to the optimal parameters. It is because the traditional algorithms do not determine the initial values by solving the model via surface data. Furthermore, the proposed metaheuristic algorithm performs exploration and exploitation, which perform a search procedure inside and outside of parents. Thus, all search space can be analyzed. This procedure leads to finding the best solution for the model parameters in an efficient form. The capability of the proposed metaheuristic genetic algorithm is elucidated by comparison with respect to the metaheuristic algorithms. To elucidate this, the quality gap, running time, and suitable structure of the metaheuristic algorithms are mentioned as follows. The particle swarm optimization provides a quality gap of 4.44% and a running time of 253 generations for spherical and cylindrical surface modeling. The ant colony optimization provides a quality gap of 4.74% and a running time of 286 generations. The simulated annealing optimization provides a quality gap of 4.91% and a running time of 292 generations. The suitable structure is compared with the particle swarm optimization that provides better results. To do so, the population evolution of the particle swarm is described as follows. The population evolution of the particle swarm is described by the expressions $V_i(t+1) = wV_i(t) + \alpha R_1[P^b{}_g(t) - P_i(t)] + \beta R_2[P^b{}_i(t) - P_i(t)]$, where $P_i(t+1) = P_i(t) + V_i(t)$, where $t$ is the number of iterations, and $\alpha$ and $\beta$ represent the learning factors. The parameters $R_1$ and $R_2$ are random numbers ranging from 0 to 1, and $w$ is the inertia weight. Based on these statements, the population of each generation is generated by a function that depends on five variables, which should be determined. The genetic algorithm determines the population of each generation by a function that depends on the variable $\beta$, which is computed via the spread factor between 0 and 1. Therefore, the genetic algorithm performs a structure that is more simple than the particle swarm optimization. In addition, ant colony optimization and simulated annealing optimization employ more variables than the genetic algorithm to construct surface models. From the results, the genetic algorithm elucidates a suitable structure to construct surface models. This statement is corroborated by the surface model constructed via genetic algorithm and by particle swarm optimization [67]. Based on these statements, the capability of the micro-scale spherical and cylindrical surface modeling via the metaheuristic algorithm based on the micro laser line projection is elucidated. Moreover, the low cost of the simple setup increases the viability of the proposed micro-scale spherical and cylindrical surface modeling. In this way, the micro-scale surface modeling via the genetic algorithm and micro laser line projection provides a contribution for the optical microscope imaging systems, which perform micro-scale surface modeling.

The computer employed to perform the micro-scale spherical and cylindrical surface modeling is a PC to 1.8 GHz of velocity. The frame rate of the camera is 58 fps. The slider device moves the microscope vision system via control software. Each micro laser line image is processed in 0.0074 s to determine the micro-scale surface contour. The spherical surface model for the plastic surface is determined in 26.21 s. The time of this procedure includes the surface scanning and the surface modeling. The cylindrical surface model for the metallic surface is determined in 25.48 s.

## 5. Conclusions

A powerful technique to perform micro-scale spherical and cylindrical surface modeling by means of a metaheuristic algorithm based on micro laser line projection with microns resolution has been presented. This spherical and cylindrical surface modeling via genetic algorithm improves the model fitting accuracy of the traditional optical microscope imaging

system, which performs surface modeling. This contribution is achieved via model fitting accuracy and genetic algorithm efficiency. The model fitting improvement is obtained by the genetic algorithm, which fits the spherical and cylindrical surface model near the surface topography. In addition, the micro-scale surface contouring via micro laser line contributes to the model fitting accuracy. It is because the micro laser line reflection reproduces the surface topography with great accuracy. Thus, the micro-scale surface model is generated from the real surface topography. The genetic algorithm efficiency is achieved by mean of the solution space, which provides analytic equations that generate initial parameters near to optimal model parameters. This leads to reducing the iterations number to find the model with the best fitting. Thus, the proposed micro-scale spherical and cylindrical surface modeling provides a valuable tool to represent spherical and cylindrical surfaces in micro-scale metrology. Additionally, the proposed technique is implemented with a simple optical hardware, which includes an optical microscope, a diode laser, and a CCD camera. Therefore, the low cost of the setup corroborates the viability of the proposed technique to perform micro-scale spherical and cylindrical surface modeling in the interval of microns. Thus, the micro-scale spherical and cylindrical surface modeling based on micro laser line projection has been performed with good results in the micro-scale interval.

This research is still working to provide some missing tools. For instance, the method just constructs spherical and cylindrical surface models. However, free-form surface models are required. In addition, the method should determine surface deformation via surface modeling. Therefore, future work is recommended as follows. Future work includes micro-scale spherical and cylindrical surface deformation modeling via metaheuristic algorithms and micro laser line scanning, micro-scale free-form surface modeling via metaheuristic algorithms and micro laser line scanning, micro-scale surface pattern characterization and recognition via algorithms of artificial intelligence and micro laser line scanning, and a micro-scale surface porosity inspection via micro laser line scanning and metaheuristic algorithms.

**Funding:** This research received no external funding.

**Institutional Review Board Statement:** Not applicable.

**Informed Consent Statement:** Not applicable.

**Data Availability Statement:** www.cio.mx (accessed on 29 December 2021).

**Conflicts of Interest:** The authors declare not conflict of interest.

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
