# Peer review of "Micro-Scale Spherical and Cylindrical Surface Modeling via Metaheuristic Algorithms and Micro Laser Line Projection"

_algorithms, doi:10.3390/a15050145_

Round 1

Reviewer 1 Report

This study focuses on micro-scale spherical and cylindrical surface modeling via metaheuristic algorithms and micro laser line projection. I think the paper fits well the scope of the journal and addresses an important subject. However, a number of revisions are required before the paper can be considered for publication. There are some weak points that have to be strengthened. Below please find more specific comments:

*Page 1 line 15: “a genetic algorithm performs the mathematical models” sounds a bit confusing. A genetic algorithm can be used to solve a mathematical model, but not to perform the model.

*The authors discuss the relevant literature in the introduction section. However, there are only 11 studies acknowledged in the introduction section. Please check for more recent and relevant works. Otherwise, the literature review may not seem convincing to some readers.

*The manuscript discusses the use of metaheuristic algorithms for surface modeling. I recommend for the authors to create a general discussion regarding the importance of advanced optimization algorithms (e.g., heuristics, metaheuristics) for challenging decision problems. There are many different domains where advanced optimization algorithms have been applied as solution approaches, such as online learning, scheduling, multi-objective optimization, transportation, medicine, data classification, and others (not just surface modeling). The authors should create a discussion that highlights the effectiveness of advanced optimization algorithms in the aforementioned domains. This discussion should be supported by the relevant references, including the following:

An online-learning-based evolutionary many-objective algorithm. Information Sciences 2020, 509, pp.1-21.

Two hybrid meta-heuristic algorithms for a dual-channel closed-loop supply chain network design problem in the tire industry under uncertainty. Advanced Engineering Informatics 2021, 50, p.101418.

A many-objective evolutionary algorithm with angle-based selection and shift-based density estimation. Information Sciences 2020, 509, pp.400-419.

An Optimization Model and Solution Algorithms for the Vehicle Routing Problem with a “Factory-in-a-Box”. IEEE Access 2020, 8, pp.134743-134763.

Exact and heuristic solution algorithms for efficient emergency evacuation in areas with vulnerable populations. International Journal of Disaster Risk Reduction 2019, 39, p.101114.

A proposal for distinguishing between bacterial and viral meningitis using genetic programming and decision trees. Soft Computing 2019, 23(22), pp.11775-11791.

Such a discussion will help improving the quality of the manuscript significantly.

*Section 2 presents micro-scale spherical surface modeling via genetic algorithm. It would be good if the authors could create a flowchart with the main steps of the algorithm.

*The manuscript contains quite a lot of figures. Please double check to make sure that all figures are adequately described in the manuscript to prevent any confusion of future readers.

*Description of the computational experiments could be improved. In particular, I recommend providing more references to support the selection of input data for the experiments.

*The discussion section seems to be adequate. No changes are needed.

*The conclusions section should expand on limitations of this study and future research needs. I suggest listing the bullet points.

Author Response

Answers for reviewer 1

Comments from the editors and reviewers:

This study focuses on micro-scale spherical and cylindrical surface modeling via metaheuristic algorithms and micro laser line projection. I think the paper fits well the scope of the journal and addresses an important subject. However, a number of revisions are required before the paper can be considered for publication. There are some weak points that have to be strengthened. Below please find more specific comments:

In reference to: 1.- Page 1 line 15: “a genetic algorithm performs the mathematical models” sounds a bit confusing. A genetic algorithm can be used to solve a mathematical model, but not to perform the model.

RE: The text has been corrected by “a genetic algorithm determines the parameters of the mathematical models to represent the spherical and cylindrical surfaces”.

In reference to: 2.- The authors discuss the relevant literature in the introduction section. However, there are only 11 studies acknowledged in the introduction section. Please check for more recent and relevant works. Otherwise, the literature review may not seem convincing to some readers.

RE: The studies acknowledged has been incremented to 42, which includes studies on surface modeling via particle swarm optimization, ant colony optimization, simulated annealing, muti-objective optimization. Also, studies on manufacturing, robotics, domains have been included.   

In reference to: 3.- The manuscript discusses the use of metaheuristic algorithms for surface modeling. I recommend for the authors to create a general discussion regarding the importance of advanced optimization algorithms (e.g., heuristics, metaheuristics) for challenging decision problems. There are many different domains where advanced optimization algorithms have been applied as solution approaches, such as online learning, scheduling, multi-objective optimization, transportation, medicine, data classification, and others (not just surface modeling). The authors should create a discussion that highlights the effectiveness of advanced optimization algorithms in the aforementioned domains. This discussion should be supported by the relevant references, including the following:

An online-learning-based evolutionary many-objective algorithm. Information Sciences 2020, 509, pp.1-21.

Two hybrid meta-heuristic algorithms for a dual-channel closed-loop supply chain network design problem in the tire industry under uncertainty. Advanced Engineering Informatics 2021, 50, p.101418.

A many-objective evolutionary algorithm with angle-based selection and shift-based density estimation. Information Sciences 2020, 509, pp.400-419.

An Optimization Model and Solution Algorithms for the Vehicle Routing Problem with a “Factory-in-a-Box”. IEEE Access 2020, 8, pp.134743-134763.

Exact and heuristic solution algorithms for efficient emergency evacuation in areas with vulnerable populations. International Journal of Disaster Risk Reduction 2019, 39, p.101114.

A proposal for distinguishing between bacterial and viral meningitis using genetic programming and decision trees. Soft Computing 2019, 23(22), pp.11775-11791.

Such a discussion will help improving the quality of the manuscript significantly.

RE: The metaheuristics algorithms construct models via multi-objective optimization in domains such as online learning, scheduling, transportation, medicine, data classification, and so on. For instance, the online learning optimizes many objective functions to determine the mathematical models [37]. Also, the multi-objective optimization via evolutionary algorithms is implemented to determine the parameters of mathematical models [38]. The other hand, the scheduling domain employs structures of different metaheuristics algorithms to determine parameters of mathematical models [39]. Also, the transportation domain employs several metaheurstics algorithms to determine routing mathematical models [40]. Additionally, the evacuation domain employs several metaheuristics algorithms to optimize the multi-objective functions to determine routing mathematical models [41]. Furthermore, the metaheuristic algorithms employ data classification to determine diagnosis in medicine [42]. The above mentioned algorithms optimize parameters for mathematical models, which are not defined by a specific equation.  Also, the metaheuristics algorithms based on multi-objective optimization determine mathematical models, which are not defined by a specific equation. This leads to implement complex algorithms due to the missing of a reference equation to optimize the model parameters. Moreover, these algorithms begin the optimization with a random solution, which leads to compute a huge iterations number. Furthermore, the objective function is deduced by an equation, which includes additional parameters to the surface model. Stead of, the spherical and cylindrical surfaces can be represented have a specific equation. In this way, a genetic algorithm provides a suitable structure to optimize the spherical and cylindrical surface models. For instance, the genetic algorithm is allowed to define an objective function by means of the equations that represent the spherical and cylindrical surface. Therefore, additional parameters are not required. Also, it is possible to begin the optimization by employing the best candidates, which provide a solution near of the optimal solution. The best candidates are deduced from the search space via spherical and cylindrical equations and known surface data. Moreover, exploration and exploitation are carried out to find the optimal solution inside or outside the best candidates. Based on these statements, the metaheuristic genetic algorithm is chosen to construct the spherical and cylindrical surface models.  Comments and references about these matters  have included in Section 1.

In reference to: 4.- Section 2 presents micro-scale spherical surface modeling via genetic algorithm. It would be good if the authors could create a flowchart with the main steps of the algorithm.

RE: The flow chart of the genetic algorithm has been included as Figure 2(b) in section 2.1.

In reference to: 5.- The manuscript contains quite a lot of figures. Please double check to make sure that all figures are adequately described in the manuscript to prevent any confusion of future readers.

RE: All Figures have been checked and described. Also, Figure 2(b) and Figure 7 have been included. 

In reference to: 6.- Description of the computational experiments could be improved. In particular, I recommend providing more references to support the selection of input data for the experiments.

RE: The first step of the genetic algorithm is to determine the initial population. Typically, the metaheuristics algorithms determine the initial population in random form. Also, the initial population can be determined by means of the best candidates. This leads to speed out the convergence [45]. The initial population plays a significant role in the genetic algorithm to provide higher qualities and better convergence [49]. The best candidates can be obtained via random methods or by sampling the surface model [50-51]. Therefore, the genetic algorithm determines the initial population by sampling the surface model to obtain the best candidates. The sampling is carried out by solving Eq.(2) and Eq.(3) by employing the surface coordinates. Comments and references about this matter are included in Section 2.2 and Section 3.

In reference to: 7.- The discussion section seems to be adequate. No changes are needed.

RE: Ok.

In reference to: 8.- The conclusions section should expand on limitations of this study and future research needs. I suggest listing the bullet points.

RE: This research is still working to some missing tools. For instance, the method just constructs spherical and cylindrical surface models. But, free-form surface models are required. Also, the method should determine surface deformation via surface modeling. Therefore, future work is pointed as follows. Future work: micro-scale spherical and cylindrical surface deformation modeling via metaheuristics algorithms and micro laser line scanning. Micro-scale free-form surface modeling via metaheuristic algorithms and micro laser line scanning. Micro-scale surface pattern characterization and recognition via algorithms of artificial intelligence and micro laser line scanning. Micro-scale surface porosity inspection via micro laser line scanning and metaheuristic algorithms.

Comments about these matters are included in the conclusions section.

Reviewer 2 Report

The author proposed a new methods for the "Micro-scale spherical and cylindrical surface modeling via metaheuristic algorithms and micro laser line projection". The method is clearly presented and the results for modelling is clearly presented. The application of metaheuristic algorithms for cylindrical surface modeling is interesting to reader. However, in order to further improve the quality of the paper. I have the following comments.

  1. The section 1.0 introduction can be extended to describe more about the past research work in these area. Furthermore, the author may include a review on past research in different metaheuristic algorithms.
  2. In the discussion section 4.0, the author could explain more about the relative merits and limitation of using metaheuristic algorithms (Genetic Algorithms) in this paper. 
  3. For section 3, the author could explain more about the reasons why GA is chosen for optimization instead of other algorithms (e.g. Particle Swarm Optimization, Simulated Annealing, Ant Colony, etc.) and describe more about the original contributions of application of GA in this problem.
  4. The author could describe more how to chosen the parameters (e.g. probability of mutation/ probability of crossover/ No. of generation etc.) for the GA.
  5. It may be useful to give a graph to show the variation of best fitness values with number of generation.
  6. Overall speaking, the quality of the paper is good and up to standard.    

Author Response

Answers for reviewer 2

Comments from the editors and reviewers:

The author proposed a new method for the "Micro-scale spherical and cylindrical surface modeling via metaheuristic algorithms and micro laser line projection". The method is clearly presented and the results for modelling is clearly presented. The application of metaheuristic algorithms for cylindrical surface modeling is interesting to reader. However, in order to further improve the quality of the paper. I have the following comments.

In reference to: 1.- The section 1.0 introduction can be extended to describe more about the past research work in these area. Furthermore, the author may include a review on past research in different metaheuristic algorithms.

RE: The recent past of the research and metaheuristics algorithms the has been included in the introduction section 1. In micro-scale manufacturing industry, object machining, surface roughness, drilling, object measurement are determined via spherical and cylindrical surface models and robotic vision systems [3-5]. To achieve these manufacturing processes, mathematical models have been implemented to represent micro-scale spherical and cylindrical surface. For instance, the spherical surface modeling has been implemented in geomatics to determine rock surface sphericity [6-7], in optics to estimate lens sphericity [8-9], in object machining to determine surface sphericity [10-11], in pharmaceutic to determine proppants sphericity [12-13], in object prototyping to determine assemble sphericity [14-15] and so on. Moreover, the cylindrical surface modeling has been implemented in surface machining to inspect cylindrical surface [16-17], in surface milling to estimate milling cylindricity [18], and so on.

The metaheuristic algorithms include algorithms such as genetic algorithms, particle swarm optimization, ant colony optimization and simulated annealing [31]. Metaheuristic algorithms such as particle swarm optimization, ant colony optimization and simulated annealing have been implemented to construct mathematical models to represent free-form surface [32-33]. For instance, particle swarm makes the optimization via position and velocity of particles [34]. Where, the objective function is provided by an inertia equation and the population is determined by a particle velocity equation. The ant colony optimization chooses paths marked by a strong pheromone concentration to determine the surface model parameters [35]. The simulated annealing makes a random search, which decreases and increases the objective function to determine the surface model [36]. Where, the tentative solution is generated by a small perturbation. Comments and references about these matters have been included in section 1.

In reference to: 2.- In the discussion section 4.0, the author could explain more about the relative merits and limitation of using metaheuristic algorithms (Genetic Algorithms) in this paper. 

RE: The contribution of the proposed technique is determined based on the model fitting to the surface. This statement includes the model fitting to the surface, optimality gap, running time, suitable structure and the algorithm efficiency. The model fitting to the surface and algorithm efficiency are achieved in good manner. The model fitting is determined via quality gap, which is computed via Eq.(27). Thus, the proposed technique fits the spherical and cylindrical surface model to the surface data with a relative error smaller than 3.8%. The algorithm efficiency is determined via structure algorithm and solution quality, which determined via running time. The Algorithm structure provides an efficient procedure to optimize the surface model parameters. It because the metaheuristic algorithm determines the initial population via best candidates, which are sampled by solving the shperical and cylindrical surface model. This procedure leads to provide initial popuation near of the optimal parameters. Thus, the algorithm provides a low error since the first generation. This leads to reduce the iterations number to obtain the optimal parameters. Therefore, the runnig time is achieved in less than 140 iterations. Thus, the proposed micro-scale surface modeling improves the model fitting of the traditional optical microscope imaging systems.  

 The capability of the proposed metaheuristic genetic algorithm is elucidated by comparison respect the metaheuristic algorithms. To elucidate this out, the quality gap, running time, and suitable structure of the metaheuristic algorithms are mentioned as follows. The particle swarm optimization provides quality gap of 4.44% and a running time of 253 generations for spherical and cylindrical surface modeling. The ant colony optimization provides quality gap of 4.74% and a running time of 286 generations. The simulated annealing optimization provides quality gap of 4.91% and a running time of 292 generations. The suitable structure is compared with the particle swarm optimization that provides better results. To do so, the population evolution of the particle swarm is described as follows. The population evolution of the particle swarm is described by the expressions Vi(t+1)=wVi(t)+ αR1[Pbg(t)Pi(t)]+βR2[Pbi(t)Pi(t)], where Pi(t+1)=Pi(t)+Vi(t). Where t is the number of iterations, and α and β represent the learning factors. The parameters R1 and R2 are random numbers ranging from 0 to 1, w is the inertia weight. Based on these statements, the population of each generation is generated by a function that depends on five variables, which should be determined. Stead of, the population of each generation is determined by a function that depends on the variable variable β, which is computed based on spread factor between 0 and 1. Therefore, the genetic algorithm performs a structure more simple than the particle swarm optimization. Also, ant colony optimization and simulated annealing optimization employ more variables than the genetic algorithm to construct surface models. From the results, the genetic algorithm elucidates a suitable structure to construct surface models. This statement is corroborated by the surface model constructed via genetic algorithm and by particle swarm optimization [67]. Based on these statements, the capability of the metaheuristic genetic algorithm is elucidated.  Comments and references are included in section 4.

In reference to: 3.- For section 3, the author could explain more about the reasons why GA is chosen for optimization instead of other algorithms (e.g. Particle Swarm Optimization, Simulated Annealing, Ant Colony, etc.) and describe more about the original contributions of application of GA in this problem.

RE: Metaheuristic algorithms such as particle swarm optimization, ant colony optimization and simulated annealing have been implemented to construct mathematical models to represent free-form surface [32-33]. The above mentioned algorithms optimize parameters for mathematical models, which are not defined by a specific equation. For instance, the particle swarm, ant colony and simulated annealing construct free-form surface models, which are not defined by a specific equation. Also, the metaheuristics algorithms based on multi-objective optimization determine mathematical models, which are not defined by a specific equation. This leads to implement complex algorithms due to the missing of a reference equation to optimize the model parameters. Moreover, these algorithms begin the optimization with a random solution, which leads to compute a huge iterations number. Furthermore, the objective function is deduced by an equation, which includes additional parameters to the surface model. Stead of, the spherical and cylindrical surfaces can be represented have a specific equation. In this way, a genetic algorithm provides a suitable structure to optimize the spherical and cylindrical surface models. For instance, the genetic algorithm is allowed to define an objective function by means of the equations that represent the spherical and cylindrical surface. Therefore, additional parameters are not required. Also, it is possible to begin the optimization by employing the best candidates, which provide a solution near of the optimal solution. The best candidates are deduced from the search space via spherical and cylindrical equations and known surface data. Moreover, exploration and exploitation are carried out to find the optimal solution inside or outside the best candidates.

Additionally, the model fitting to the surface, and the algorithm efficiency are evaluated to elucidate the capability of the proposed algorithm. The model fitting to the surface and algorithm efficiency are achieved in good manner. The model fitting is determined via quality gap, which is computed via Eq.(27). Thus, the proposed technique fits the spherical and cylindrical surface model to the surface data with a relative error smaller than 3.8%. The algorithm efficiency is determined via structure algorithm and solution quality, which determined via running time. The Algorithm structure provides an efficient procedure to optimize the surface model parameters.  

The capability of the proposed metaheuristic genetic algorithm is elucidated by comparison respect the metaheuristic algorithms. To elucidate this out, the quality gap, running time, and suitable structure of the metaheuristic algorithms are mentioned as follows. The particle swarm optimization provides quality gap of 4.44% and a running time of 253 generations for spherical and cylindrical surface modeling. The ant colony optimization provides quality gap of 4.74% and a running time of 286 generations. The simulated annealing optimization provides quality gap of 4.91% and a running time of 292 generations. The suitable structure is compared with the particle swarm optimization that provides better results. To do so, the population evolution of the particle swarm is described as follows. The population evolution of the particle swarm is described by the expressions Vi(t+1)=wVi(t)+ αR1[Pbg(t)Pi(t)]+βR2[Pbi(t)Pi(t)], where Pi(t+1)=Pi(t)+Vi(t). Where t is the number of iterations, and α and β represent the learning factors. The parameters R1 and R2 are random numbers ranging from 0 to 1, w is the inertia weight. Based on these statements, the population of each generation is generated by a function that depends on five variables, which should be determined. Stead of, the population of each generation is determined by a function that on the variable variable β, which is computed based on value between 0 and 1. Therefore, the genetic algorithm performs a structure more simple than the particle swarm optimization. Also, ant colony optimization and simulated annealing optimization employ more variables than the genetic algorithm to construct surface models. From the results, the genetic algorithm elucidates a suitable structure to construct surface models. This statement is corroborated by the surface model constructed via genetic algorithm and by particle swarm optimization [67]. Based on these statements, the capability of the metaheuristic genetic algorithm is elucidated.  Comments and references about these matters are included in section 1 and section 4.

In reference to: 4.- The author could describe more how to chosen the parameters (e.g. probability of mutation/ probability of crossover/ No. of generation etc.) for the GA.

RE: The probability of crossover is determined via fitness [52]. When the average fitness of the parents is improved, the crossover is carried out. This procedure leads to prevent the loss of candidates to achieve the convergence. Thus, the probability of crossover is in the interval from 0.0 to 0.5. This result is obtained by computing several times the same experiment.

The mutation probability is determined via fitness. For the mutation of the parent, if the new parent improves the fitness, the worst parent is mutated. The other hand, the worst parent is not mutated. For the parameter mutation, if the new parameter improves the parent fitness, the parameter is mutated. Thus, the mutation probability crossover is in the interval from 0.0 to 0.62. This result is obtained by computing several times the same experiment.

The number of generations is deduced from the iterations number to obtain the optimized parameters.

Comments and references about these matters are included in section 3.

In reference to: 5.- It may be useful to give a graph to show the variation of best fitness values with number of generation.

RE: The fitness variation with respect to the generation number is depicted by the graphic shown in Figure 7.

In reference to: 6.- Overall speaking, the quality of the paper is good and up to standard.    

RE: Ok.

Reviewer 3 Report

Manuscript Number: algorithms-1557772-peer-review-v1

Title: Micro-scale spherical and cylindrical surface modeling via metaheuristic algorithms and micro laser line projection

This study focuses on metaheuristic algorithms based on micro laser line projection to perform micro-scale spherical and cylindrical surface modeling. As authors mentioned, a genetic algorithm (GA) performs the mathematical models from the surface coordinates to represent the spherical and cylindrical surfaces. The GA performs exploration and exploitation in the search space to optimize the model mathematical parameters.

This paper requires a revision to reach a satisfactory quality in the problem definition, mathematical modeling, solution method and outcomes.

- The main issue is that authors did not clarify why they are using metaheuristics, and particularly GA, for the problem? Why this is better than other approaches? Is it about search space or NP-hardness of the problem? Regarding GA, for example, I know it is good at taking large, potentially huge search spaces and navigating solutions to look for optimal combinations of solutions you might not otherwise find in a lifetime. When the problem is NP-hard, it has ability to locate the neighborhood of the optimal solution quicker than other conventional search strategies.

Authors should explain why they have selected this method.

-I can see that authors put a considerable effort into enhancing the paper. I wonder how they could evaluate the performance of the GA via optimality gap, solution (running time) quality, etc. to justify the performance of the algorithm.

-Another main issue is the organization of the paper. The term “Manufacturing” is a keyword that is only in abstract. There is not much about manufacturing industry and their application mentioned in the body of the paper, but I think it should be more detailed in the body of the paper, at least in the introduction section. Authors should add more about manufacturing and industry applications to help readers know why the paper is important.

- In Section 1, I have found that robotics is a good application area. I suggest citing recent shop floor studies using sensor, camera, and laser to improve manufacturing system productivity and product quality. [a] Automatic self-contained calibration of an industrial dual-arm robot with cameras using self-contact, planar constraints, and self-observation, Robotics and Computer-Integrated Manufacturing, vol 73, 102250 [b] Stochastic optimization of two-machine flow shop robotic cells with controllable inspection times: From theory toward practice, Robotics and Computer-Integrated Manufacturing, vol. 61, 101822.

- There is no future research direction in the last section. Please add a couple of future research directions. It should include (1) building on a particular finding in your research; (2) addressing a flaw in your (future) research; examining (or testing) a theory (framework or model) either, etc.

-Some words should be capitalized. Examples are:

Page 2: in section 2.1--> in Section 2.1

Page 2: in section 2.2 --> in Section 2.2

-I can see both “Figure” and “Fig.” in the body of the paper. This is an inconsistency. Please use one of them, not both.

- Other errors:

Page 3: where the sub-indexes --> where the sub-indices

Page 4: via sub-indexes --> via sub-indices

Note: Indices is generally preferred in mathematical, financial, and technical contexts, while indexes is relatively common in general usage.

Page 4: implemented via Eq.(1), Eq.(2) and Eq.(3) --> implemented via Eqs.(1-3)

Page 14: data over surface surface model --> data over surface model

The paper benefits from proofread.

Author Response

Answers for reviewer 3

Comments from the editors and reviewers:

Reviewer 3

This study focuses on metaheuristic algorithms based on micro laser line projection to perform micro-scale spherical and cylindrical surface modeling. As authors mentioned, a genetic algorithm (GA) performs the mathematical models from the surface coordinates to represent the spherical and cylindrical surfaces. The GA performs exploration and exploitation in the search space to optimize the model mathematical parameters.

This paper requires a revision to reach a satisfactory quality in the problem definition, mathematical modeling, solution method and outcomes.

In reference to: 1.- The main issue is that authors did not clarify why they are using metaheuristics, and particularly GA, for the problem? Why this is better than other approaches? Is it about search space or NP-hardness of the problem? Regarding GA, for example, I know it is good at taking large, potentially huge search spaces and navigating solutions to look for optimal combinations of solutions you might not otherwise find in a lifetime. When the problem is NP-hard, it has ability to locate the neighborhood of the optimal solution quicker than other conventional search strategies.

Authors should explain why they have selected this method.

RE: The justification to select the metaheuritic GA is based on the parameters such as optimality gap, running time, and suitable structure. The metaheuristic algorithms include algorithms such as genetic algorithms, particle swarm optimization, ant colony optimization and simulated annealing [31]. Metaheuristic algorithms such as particle swarm optimization, ant colony optimization and simulated annealing have been implemented to construct mathematical models to represent free-form surface [32-33]. For instance, particle swarm makes the optimization via position and velocity of particles [34].  The ant colony optimization chooses paths marked by a strong pheromone concentration to determine the surface model parameters [35]. The simulated annealing makes a random search, which decreases and increases the objective function to determine the surface model [36]. Also, the multi-objective optimization via evolutionary algorithms is implemented to determine the parameters of mathematical models [38]. The above mentioned algorithms optimize parameters for mathematical models, which are not defined by a specific equation. This leads to implement complex algorithms due to the missing of a reference equation to optimize the model parameters. Moreover, these algorithms begin the optimization with a random solution, which leads to compute a huge iterations number. Furthermore, the objective function is deduced by an equation, which includes additional parameters to the surface model. Stead of, the spherical and cylindrical surfaces can be represented have a specific equation. In this way, a genetic algorithm provides a suitable structure to optimize the spherical and cylindrical surface models. For instance, the genetic algorithm is allowed to define an objective function by means of the equations that represent the spherical and cylindrical surface. Therefore, additional parameters are not required. Also, the initial population determined by employing the best candidates, which provide a solution near of the optimal solution. This leads to speed out the convergence. The best candidates are deduced from the search space via spherical and cylindrical equations and known surface data.

Moreover, exploration and exploitation are carried out to find the optimal solution inside or outside the best candidates. This takes large, potentially huge search spaces and navigating solutions to look for optimal combinations of solutions. Based on these statements, the metaheuristic genetic algorithm is chosen to construct the spherical and cylindrical surface models. The genetic algorithm viability to construct micro-scale spherical and cylindrical surface models is elucidated via quality gap, solution quality and experimental results.

The capability of method is established based on the model fitting accuracy [55-56]. This statement includes the model fitting to the surface, and the algorithm efficiency. The model fitting to the surface and algorithm efficiency are achieved in good manner. The model fitting is determined via quality gap, which is computed via Eq.(27). Thus, the proposed technique fits the spherical and cylindrical surface model to the surface data with a relative error smaller than 3.8%. The algorithm efficiency is determined via structure algorithm and solution quality, which determined via running time. The Algorithm structure provides an efficient procedure to optimize the surface model parameters. It because the metaheuristic algorithm determines the initial population via best candidates, which are sampled by solving the shperical and cylindrical surface model. This procedure leads to provide initial popuation near of the optimal parameters.  Thus, the algorithm provides a low error since the first generation. This leads to reduce the iterations number to obtain the optimal parameters. Therefore, the runnig time is achieved in less than 140 iterations.  

Additionally, the capability of the proposed metaheuristic genetic algorithm is elucidated by comparison respect the metaheuristic algorithms. To elucidate this out, the quality gap, running time, and suitable structure of the metaheuristic algorithms are mentioned as follows. The particle swarm optimization provides quality gap of 4.44% and a running time of 253 generations for spherical and cylindrical surface modeling. The ant colony optimization provides quality gap of 4.74% and a running time of 286 generations. The simulated annealing optimization provides quality gap of 4.91% and a running time of 292 generations. The suitable structure is compared with the particle swarm optimization that provides better results. To do so, the population evolution of the particle swarm is described as follows. The population evolution of the particle swarm is determined by the expressions Vi(t+1)=wVi(t)+αR1[Pbg(t)Pi(t)]+βR2[Pbi(t)Pi(t)], where Pi(t+1)=Pi(t)+Vi(t). Where t is the number of iterations, and α and β represent the learning factors. The parameters R1 and R2 are random numbers ranging from 0 to 1, w is the inertia weight. Based on these statements, the population of each generation is generated by a function that depends on five variables, which should be determined. Stead of, the population of each generation is determined by a function that depends on the variable variable β, which is computed based on a spread factor between 0 and 1. Therefore, the genetic algorithm performs a structure more simple than the particle swarm optimization. Also, ant colony optimization and simulated annealing optimization employ more variables than the genetic algorithm to construct surface models. From the results, the genetic algorithm elucidates a suitable structure to construct surface models. This statement is corroborated by the surface model constructed via genetic algorithm and by particle swarm optimization [67]. Based on these statements, the capability of the metaheuristic genetic algorithm is elucidated.  Comments and references are included in Section 1 and Section 4.

In reference to: 2.- I can see that authors put a considerable effort into enhancing the paper. I wonder how they could evaluate the performance of the GA via optimality gap, solution (running time) quality, etc. to justify the performance of the algorithm.

RE: The capability of method is established based on the model fitting accuracy [55-56]. This statement includes the model fitting to the surface, and the algorithm efficiency. The model fitting to the surface and algorithm efficiency are achieved in good manner. The model fitting is determined via quality gap, which is computed via Eq.(27). Thus, the proposed technique fits the spherical and cylindrical surface model to the surface data with a relative error smaller than 3.8%. The algorithm efficiency is determined via structure algorithm and solution quality, which determined via running time. The Algorithm structure provides an efficient procedure to optimize the surface model parameters. It because the metaheuristic algorithm determines the initial population via best candidates, which are sampled by solving the shperical and cylindrical surface model. This procedure leads to provide initial popuation near of the optimal parameters.  Thus, the algorithm provides a low error since the first generation. This leads to reduce the iterations number to obtain the optimal parameters. Therefore, the runnig time is achieved in less than 140 iterations.  Comments and references about these matters are included in Section 3 and Section 4.

In reference to: 3.- Another main issue is the organization of the paper. The term “Manufacturing” is a keyword that is only in abstract. There is not much about manufacturing industry and their application mentioned in the body of the paper, but I think it should be more detailed in the body of the paper, at least in the introduction section. Authors should add more about manufacturing and industry applications to help readers know why the paper is important.

RE: In micro-scale manufacturing industry, object machining, surface roughness, drilling, object measurements are determined via spherical and cylindrical surface models and robotic vision systems [3-5]. To achieve these manufacturing processes, mathematical models have been implemented to represent micro-scale spherical and cylindrical surface.  For instance, the spherical surface modeling has been implemented in geomatics to determine rock surface sphericity [6-7], in optics to estimate lens sphericity [8-9], in object machining to determine surface sphericity [10-11], in pharmaceutic to determine proppants sphericity [12-13], in object prototyping to determine assemble sphericity [14-15] and so on. Moreover, the cylindrical surface modeling has been implemented in surface machining to inspect cylindrical surface [16-17], in surface milling to estimate milling cylindricity [18], and so on. Comments and references about this matter are included in Section 1.

In reference to: 4.- In Section 1, I have found that robotics is a good application area. I suggest citing recent shop floor studies using sensor, camera, and laser to improve manufacturing system productivity and product quality. [a] Automatic self-contained calibration of an industrial dual-arm robot with cameras using self-contact, planar constraints, and self-observation, Robotics and Computer-Integrated Manufacturing, vol 73, 102250 [b] Stochastic optimization of two-machine flow shop robotic cells with controllable inspection times: From theory toward practice, Robotics and Computer-Integrated Manufacturing, vol. 61, 101822.

RE: These references [5], [25] have been included in section 1.

In reference to: 5.- There is no future research direction in the last section. Please add a couple of future research directions. It should include (1) building on a particular finding in your research; (2) addressing a flaw in your (future) research; examining (or testing) a theory (framework or model) either, etc.

RE: Future work: micro-scale spherical and cylindrical surface deformation modeling via metaheuristics algorithms and micro laser line scanning. Micro-scale free-form surface modeling via metaheuristic algorithms and micro laser line scanning. Micro-scale surface pattern characterization and recognition via algorithms of artificial intelligence and micro laser line scanning. Micro-scale surface porosity inspection via micro laser line scanning and metaheuristic algorithms. Comments and references about this matter have been  included in Section of conclusions.

6.-Some words should be capitalized. Examples are:

In reference to: Page 2: in section 2.1--> in Section 2.1

RE:  “in section 2.1” has been changed by “in Section 2.1”

In reference to: Page 2: in section 2.2 --> in Section 2.2

RE: “in section 2.2” has been changed by “in Section 2.2”

In reference to: 7.- I can see both “Figure” and “Fig.” in the body of the paper. This is an inconsistency. Please use one of them, not both.

RE: “Fig.” has been changed by Figure.

8.- Other errors:

In reference to: Page 3: where the sub-indexes --> where the sub-indices

RE: “sub-indexes” has been changed by sub-indices.

In reference to: Page 4: via sub-indexes --> via sub-indices

RE: “sub-indexes” has been changed by sub-indices.

In reference to: Page 4: implemented via Eq.(1), Eq.(2) and Eq.(3) --> implemented via Eqs.(1-3)

RE: “implemented via Eq.(1), Eq.(2) and Eq.(3)” has been changed by “implemented via Eqs.(1-3)”.

In reference to: Page 14: data over surface surface model --> data over surface model

RE: “data over surface surface model” has been changed by “ data over surface model”.

In reference to: The paper benefits from proofread.

RE: Thanks.

Round 2

Reviewer 1 Report

The authors took seriously my previous comments and made the required revisions in the manuscript. The quality and presentation of the manuscript have been improved. Therefore, I recommend acceptance.

Reviewer 3 Report

I have read the paper once more to see if my comments are resolved. 

The author could significantly improve the quality of the paper. Almost all my comments are responded properly. So, I believe the paper can be accepted as it is.